# Memory by accident: a theory of learning as a byproduct of network stabilization

**Basile Confavreux**
Gatsby Computational Neuroscience Unit
University College London
basile.confavreux@gmail.com

**William Dorrell**
Gatsby Computational Neuroscience Unit
University College London

**Nishil Patel**
Gatsby Computational Neuroscience Unit
University College London

**Andrew Saxe**
Gatsby Computational Neuroscience Unit
University College London

## Abstract

Synaptic plasticity is widely considered to be crucial to the brain's ability to learn throughout life. Decades of theoretical work have therefore been invested in deriving and designing biologically plausible learning rules capable of granting various memory abilities to neural networks. Most of these theoretical approaches optimize directly for a desired memory function; but this procedure can lead to complex, finely-tuned rules, rendering them brittle to perturbations and difficult to implement in practice. Instead, we build on recent work that automatically discovers large numbers of candidate plasticity rules operating in recurrent spiking neural networks. Surprisingly, despite the fact that these rules are selected solely to achieve network stabilization, we observe across a range of network models—feedforward, recurrent; rate and spiking—that almost all these rules endow the network with simple forms of memory such as familiarity detection - seemingly by accident. To understand this phenomenon, we study an analytic toy model. We observe that memory arises from the degeneracy of weight matrices that stabilize a network: where the network lands in this space of stable weights depends on its past inputs—that is, memory. Even simple Hebbian plasticity rules can utilize this degeneracy, creating a zoo of memory abilities with various lifetimes. In practice, the larger the network and the more co-active plasticity rules in the system, the stronger the memory-by-accident phenomenon becomes. Overall, our findings suggest that activity-silent memory is a near-unavoidable consequence of stabilization. Simple forms of memory, such as familiarity or novelty detection, appear to be widely available resources for plastic brain networks, suggesting that they could form the raw materials that were later sculpted into higher-order cognitive abilities.

## 1 Introduction

Understanding how the brain orchestrates its plastic synapses is a holy grail of neuroscience, a potential stepping stone on the way to generalist models of brain function. Yet, despite concerted effort, empirical plasticity data remains scarce, largely due to the difficulty of recording synapses *in vivo* during learning. Consequently, theory and computation have played an out-sized role in developing and testing plasticity hypotheses [1]. Most theories derive their proposed learning rules by optimizing for some reasonable neural goals; for a memory system these might include the capacity, reliability or decodability of memories, while enforcing biological plausibility constraints such as

39th Conference on Neural Information Processing Systems (NeurIPS 2025).

locality or slowly changing weights [2–5]. Rules so derived are promising candidates, but their fine-tuning to a specific function can make them brittle to perturbations and hard to implement in practice [6–8]. Besides, though frameworks analyzing the learning dynamics of spike-timing dependent plasticity rules at steady state in spiking networks exist—mainly mean-field methods [9–11]—it remains challenging to capture analytically the interactions between co-active plasticity rules operating at different synapse types in parallel, restricting most theories to study rules in isolation.

Instead, recent work has used numerical optimization to propose candidate learning rules, via meta-learning techniques [12–19]. This has led to the discovery of entire families of novel co-active learning rules able to robustly stabilize large recurrent spiking networks [19, 20]. Interestingly, despite being selected *only* to ensure that the spiking network stays stable, the majority of rules display a range of interesting memory behaviors, such as novelty detection, contextual novelty and replay [20]. It seems that basic memory abilities are a natural byproduct of network stabilization.

Here we seek to understand this link between stability and memory. By **stability**, we mean networks with "biologically plausible" activity and weight dynamics over a wide range of inputs - i.e. plausible asynchronous neural activities and slowly changing weights bounded within reasonable ranges - as in previous work [20]. We employ a broader definition of **memory** in this paper than, for example, associative memories in Hopfield networks [21]. Instead, we focus on familiarity/novelty detection: "the ability to discriminate between the relative familiarity or novelty of stimuli" [22], which has a long history in psychology, experimental and computational neuroscience [23, 22, 24]. This form of memory is simpler, and can exist without, for instance, pattern completion or an attractor state for each memory.

We study a range of network models—from large recurrent spiking to shallow linear feedforward rate networks—and show that the link is robust: across models, learning rules built to stabilize neural activity consistently encode memories with various lifetimes and properties. By reverse engineering spiking networks and studying analytic toy models, we show that these memories have a simple origin: for any input there is a degeneracy of stable weight matrices. Which matrix the learning rules select often depends on the network's history, providing the basis for memory storage and recall. Large-scale simulations suggest that the bigger the system and the more co-active learning rules, the longer-lasting and more robust the memories. The remarkable ease with which such memory abilities arise in stable spiking networks may form the basis of higher-order cognitive abilities as compositions of simpler, ubiquitous, memorization skills. As such, we present these ideas as a fresh take on the emergence of memory in the brain: memory by *accident*.

## 2 Memory is a common byproduct of stabilization by co-active plasticity rules

Our starting point was a manifold of co-active synaptic plasticity rules that enforce stable dynamics in large recurrent spiking networks. There are four types of synapses in these networks —Excitatory (E)-to-E, E-to-Inhibitory (I), I-to-E and I-to-I— each governed by its own plasticity rule. The rules are parameterized by their dependence on the pre-synaptic spike train, post-synaptic spike train and a Hebbian term dependent on both (fig. 1A [19, 20]):

$$\frac{dw(t)}{dt} = \eta[\underbrace{S_{\text{pre}}(t)\big(\alpha_{\text{XY}} + \underbrace{\kappa_{\text{XY}}x_{\text{post}}(t)}_{\text{Hebbian}}\big)}_{\text{Pre-synaptic}} + \underbrace{S_{\text{post}}(t)\big(\beta_{\text{XY}} + \underbrace{\gamma_{\text{XY}}x_{\text{pre}}(t)}_{\text{Hebbian}}\big)}_{\text{Post-synaptic}}], \ \ \text{X,Y} \in \{\text{E,I}\} \quad (1)$$

where $S_{\text{pre}}(t)$ and $S_{\text{post}}(t)$ are spike trains, $x_{\text{pre}}(t)$ and $x_{\text{post}}(t)$ are their low-pass filtered versions. In sum, for each plasticity rule, there are 6 parameters, four as in eq. (1) ($\alpha, \beta, \gamma, \kappa$), and two timescales ($\tau^{\text{pre}}, \tau^{\text{post}}$), one for each low-pass filtering. Recent work meta-learned thousands of choices of these parameters that led to networks with stable dynamics – meaning the activities and weights in the networks remained in plausible ranges across many inputs for at least 4 hours [20]. This led to a "stability manifold", a stable subset of the 24 dimensional plasticity parameter space.

Despite selecting these rule quadruplets for stability, previous work observed that simple forms of memory were a near ubiquitous byproduct of network stabilization [20]. For example, the networks were tested on a familiarity detection task (fig. 1A): during a training period the networks were presented with a subset of the stimuli, then, after a variable time period, the network weights were frozen and the network was presented with both novel and familiar stimuli. The network was said to remember the familiar stimuli if the average firing rate was significantly different between the familiar and novel stimuli presentations. In this task almost all the rules showed some form of memory, with lifetimes ranging from seconds to hours (fig. 1B). In other words, most rules created memory traces

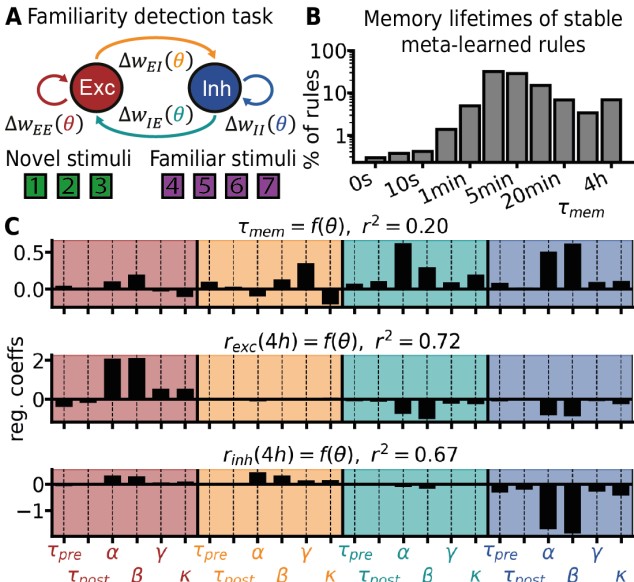

Figure 1: **A**: Recurrent spiking network ($N_{\text{E}} = 4096$, $N_{\text{I}} = 1024$) with four co-active plasticity rules undergoing a familiarity detection task. **B**: Distribution of elicited memory lifetime in the familiarity task among 2500 stable meta-learned rule quadruplets. Memory lifetime was defined as the last time point at which the network population firing rate was significantly different during novel vs familiar stimulus presentation (Student t-test, $p < 0.05$ for $N_{\text{trials}} = 5$). Original data from [20]. **C**: Linear regressions, predicting from the values of the 24 plasticity parameters: the memory lifetime of each quadruplet on the familiarity task (top); the mean firing rate of the excitatory population after 4h (middle); the mean inhibitory firing rate after 4h (bottom). Dataset of 2500 rule quadruplets taken from [20]. While mean rates can be readily predicted from plasticity, memory lifetimes cannot.

with behaviorally-relevant lifetimes from a single stimulus presentation, despite having Hebbian pairing windows of tens to hundreds of milliseconds.

We wondered which features of the plasticity rules determined their memory abilities. Simply linearly predicting the memory lifetime from the plasticity parameters was impossible (fig. 1C) while, as a control, it was possible to predict other, more basic, network properties such as the excitatory and inhibitory mean firing rates using this strategy (fig. 1C). To understand this memory-by-accident phenomenon, we therefore take a different approach. In the following sections, we reverse engineer these memories across a range of models, including a simple analytic toy model, before returning to test the intuitions we develop on the full co-active spiking networks in the final section.

## 3 Reverse engineering memory by accident in spiking networks

### 3.1 Recurrent spiking network with four co-active rules

First, we investigated how the simple co-active rules from the stability manifold were both creating long-lived memories and enforcing stability in recurrent spiking networks. We focused on one rule quadruplet which responded significantly to the familiarity task for over 4h (fig. 2A). This quadruplet stabilized the network by driving it to an activity setpoint at around 2Hz during a pre-training phase of random background inputs (fig. 2A). The network was then perturbed by the input of the (not yet) familiar patterns, though the activity recovered to its pre-training setpoint less than a minute later (fig. 2A). However, probing the network with either familiar or novel stimuli elicited responses of different magnitudes (fig. 2A and fig. S2), and, as advertised, these differences persisted over the remaining hour of simulation. The meta-learned quadruplet thus elicited a form of long-lasting, activity-silent memory [25]. Further, unlike the network activity, neither the mean or variance of the weights recovered to their pre-training values (fig. 2A). This indicated that the weight matrices had experienced long-lasting changes that enabled the different network responses observed for familiar vs novel stimuli.

To analyze the weights in this network we defined the "engram" for each stimulus as the neurons with the highest $10\%$ of activities in response to the stimuli. Since, by construction, even the neurons in naive networks were tuned to particular inputs (see fig. S2), this engram was meaningful even during pre-training phases. We observed that training mainly affected the weights to and from the familiar stimuli's engrams (fig. 2B, time point 2, strengthening of $W_{IE}$ engram weights, and weakening of $W_{EE}$ engram weights), and these familiar-specific changes persisted long after the end of training (fig. 2B, time point 3). We hypothesize that it is these familiar-specific weight changes that enabled long-term memory recall. In other words, among the many $(W_{EE}, W_{EI}, W_{IE}, W_{II})$ matrices able to stabilize the network in background state (the degeneracy of solutions to the stabilization problem), the specific weight matrices reached by plasticity reflected the system's history—here past input stimuli—giving rise to memory.

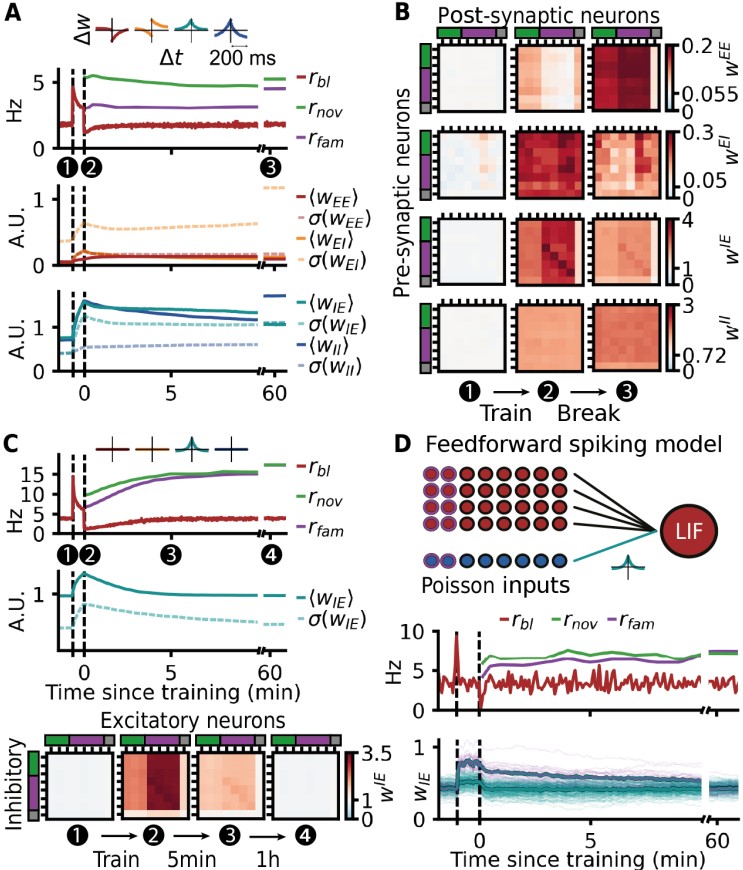

Figure 2: **A**: Example meta-learned rule quadruplet undergoing the familiarity task. Top: visualization of the rule quadruplet - normalised weight changes elicited by a pair of pre-synaptic and post-synaptic spikes with different time-lag between spikes. Second plot: excitatory population firing rate without any active stimuli (red, baseline) during the task, dashed lines denote the start and end of training. After training, average excitatory population firing rate in response to four novel (green) and three familiar stimuli (purple). Third & Fourth plot: Evolution of the mean and standard deviation of the weights during the task. **B**: Weight matrices of the network for each synapse type, at three time points: (1) immediately before the start of training, (2) immediately after training, (3) one hour after the end of training. We sort the weights according to neuron response class: green - neurons responding to novel patterns; purple - neurons responding to familiar patterns (shown during training); grey - all other neurons. **C**: Same as A&B, but for a recurrent spiking network with iSTDP (I-to-E only, [11]). **D**: Feedforward spiking network model. Excitatory and inhibitory Poisson inputs project on a single output neuron. The familiarity task followed a similar structure to the recurrent case. Middle: output neuron firing rate. Bottom: evolution of the 200 inhibitory (plastic) weights. Weights from input neurons belonging to the familiar stimulus are shown in purple. Means of the two groups ("familiar" weights vs rest) are in bold.

## 3.2 Recurrent spiking network with only I-to-E plasticity

We next sought to test our intuitions in a more tractable setting - recurrent spiking networks with a single operational plasticity rule. Within the stability manifold we recognized iSTDP, a widely-used rule for stabilising spiking networks [11, 26, 27]. This is a symmetric Hebbian rule operating only on the I-to-E connectivity that has been studied analytically (fig. 2C). Networks operating under this rule are known to settle to a stable mean firing rate whose value can be calculated using a mean-field approach [11].

We simulated networks evolving under iSTDP and observed very similar patterns to that of the long-term memory quadruplet: the activity settled both before and after training to a setpoint (3Hz). Further, despite showing no activity-related memory traces, an imprint of the familiar stimuli could be seen in the weights (fig. 2C), though in this case the memories lasted only a few minutes. It seemed that the network exhibited a separation of timescales: fast dynamics that restored the activity to the set point, and slower dynamics that erased the imprint of the memory from the connectivity weights (fig. 2C). This separation led to a period of time in which activity was flat yet the network was able to produce reliable memory-by-accident, matching our findings in the quadruplet rule case.

## 3.3 Feedforward spiking network with I-to-E plasticity

Finally, we study one further simplified spiking network: a feedforward network undergoing the familiarity detection task (fig. 2D). This model comprised a single leaky-integrate-and-fire output neuron receiving inputs from 800 excitatory and 200 inhibitory Poisson neurons. Only the inhibitory weights were plastic, following the iSTDP rule [11]. Initially, all input neurons fired at the same rate, then we performed stimulus presentation by elevating the activity of 100 excitatory and 25 inhibitory input neurons - the "engram neurons" in this setting (see Supplementary for more details).

Once again, transient memorization of the familiar stimulus could be seen (fig. 2D). Matching our observations in recurrent networks, we observed that weights from inhibitory engram neurons relaxed towards the background weight distribution at two different timescales after stimulus presentation. Fast dynamics with a time constant of a few seconds returned the output neuron's firing rate to the firing rate set point; while a slower relaxation, lasting from minutes to hours, erased the imprint of the memory from the weights. Since the only force that drove weight change in this network was deviation of the output neuron's firing rate from a target, these slower changes seemed to be driven by random fluctuations that slowly overshadowed the memory, leading to observable memory-by-accident phenomena persisting for behavioral timescales. We also verified that this insight was not specific to I-to-E plasticity by running a similar network and task with an E-to-E rule instead (fig. S5).

Overall, our analysis of each plastic spiking network confirmed the idea that the degeneracy of weight configurations capable of stabilizing the network provided the substrate for the formation of "accidental" memories. To understand this precisely we now turn to some analytic toy models.

## 4 Building a toy model for memory by accident

### 4.1 Explicit feedforward model

So far we have analyzed a few networks, each following a single set of co-active plasticity rules. However, these simulations do not help us to understand the ubiquity of memory-by-accident across the set of all stabilizing plasticity rules. To that end, we build an analytic toy model from which we can derive the memory properties of networks following various stabilizing plasticity rules. We thus turn to a minimal firing rate model capable of self-stabilization.

We considered a linear feedforward network with two inputs $\boldsymbol{x} = (x_0, x_1)$ and a single output $y$: $y(t) = w_0(t)x_0(t) + w_1(t)x_1(t)$ (fig. 3). All activities were nonnegative (though weights were unconstrained), and for simplicity we made the inputs unit norm. We considered a four-parameter set of Hebbian/non-Hebbian plasticity rules inspired by the full spiking model:

$$\frac{\partial w_i(t)}{\partial t} = \theta_0 + \theta_1 x_i(t) + \theta_2 y(t) + \theta_3 x_i(t)y(t) \tag{2}$$

Not all these plasticity rules are meaningful, we therefore restricted to rules that produced stable output activity for all possible inputs, a redefinition of *stability* for this simple feedforward model. In

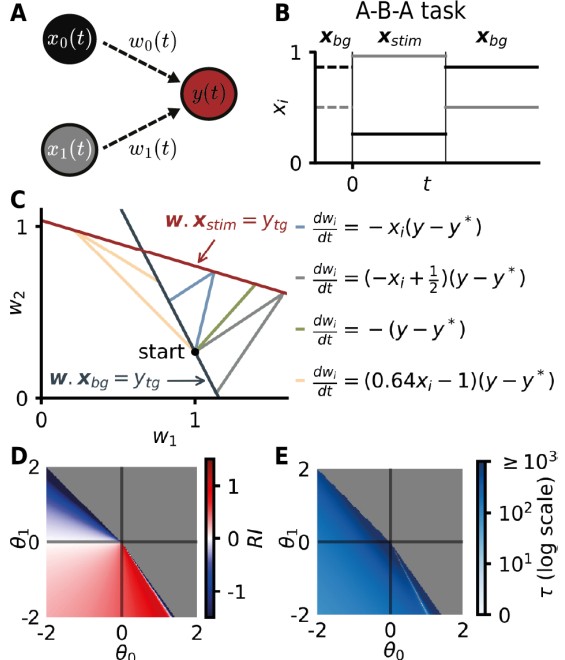

Figure 3: **A**: Feedforward linear network with two input neurons and one output neuron. **B**: Simplified version of the familiarity task. Starting from a steady state solution for $x_{\mathrm{bg}}$, the network was shown $x_{\mathrm{stim}}$ until convergence then $x_{\mathrm{bg}}$ until convergence. **C**: Dynamics of four learning rules in the A-B-A task, shown in weight space. All networks were initialized at the same state $w_0 \in W^*_{x_{\mathrm{bg}}}$ denoted as "start". **D**: Phase portrait of the relative improvement $RI$ as a function of the values of $\theta_0$ and $\theta_1$. This was computed for $\Theta_{x_{\mathrm{bg}}} = \frac{\pi}{6}$ and $\Theta_{x_{\mathrm{stim}}} = \frac{\pi}{6} + \frac{\pi}{4}$. Grey denotes unstable rules. **E**: Same parameter sweep as C, but plotting the time to convergence $\tau$.

particular, we chose a target output firing rate, $y^*$, and derived constraints on the plasticity parameters $\{\theta_j\}_{0 \leq j \leq 3}$ such that the output activity eventually converged to $y^*$ no matter which unit norm input was presented (see Supplementary for derivation). These assumptions reduced the set of feasible learning rules to the following two-parameter model:

$$\frac{\partial w_i(t)}{\partial t} = (y(t) - y^*)(\theta_0 + \theta_1 x_i(t)), \ s.t. \ \theta_0 + \theta_1 < 0 \text{ and } \sqrt{2}\theta_0 + \theta_1 < 0 \qquad (3)$$

In this setting we devised a simplified task to assess memory, the A-B-A task (fig. 3B). The network received two stimuli, $x_{\mathrm{bg}}$ and $x_{\mathrm{stim}}$, both unit norm and aligned at a random, smaller than 90°, angle from the x axis. For each input there is a stable subset of weight configurations, and in this case, due to the linearity of the problem, they form a line. We define $W^*_x = \{w, w.x = y^*\}$ as the line of weight vectors that produce a stable output firing rate ($y = y^*$) for a given input $x$. We initialized the network at $w_0 \in W^*_{x_{bg}}$, i.e. at a stable point for the background input $x_{\mathrm{bg}}$. We then presented the stimulus, ran the network to convergence, before returning again to the background input and running to convergence (fig. 3C). In this simple setting, we defined the "memory" of the system in two ways. First via the relative improvement RI:

$$\mathrm{RI}(\theta_0, \theta_1, y^*) = \frac{(w_{\mathrm{bg}'} - w_{\mathrm{bg}}) . (w_{\mathrm{bg} \cap \mathrm{stim}} - w_{\mathrm{bg}})}{||w_{\mathrm{bg} \cap \mathrm{stim}} - w_{\mathrm{bg}}||^2} \qquad (4)$$

$w_{\mathrm{bg}}$ denotes the steady state weights for input $x_{\mathrm{bg}}$ at the start of the task , while $w_{\mathrm{bg}'}$ are those for $x_{\mathrm{bg}}$ at the end of the task. $w_{\mathrm{bg} \cap \mathrm{stim}}$ is the intersection between $W^*_{x_{\mathrm{bg}}}$ and $W^*_{x_{\mathrm{stim}}}$. The magnitude of RI encodes the distance between the initial and final state of the network, while its sign indicates whether the final state was closer to the intersection between the two lines of fixed points or not. Zero indicates that the final weights are identical to the initial weights, while positive (negative) values indicate net movement towards (away from) the weight matrix that stabilizes both $x_{\mathrm{bg}}$ and $x_{\mathrm{stim}}$. Second, we evaluated memory with the time to convergence to the fixed point $\tau(\theta_0, \theta_1, y^*) = \min(t, |y(t) - y^*| \leq \rho)$, with $\rho = 0.01$ a threshold.

Almost all rules exhibited some form of memory in this toy model, i.e. the final and initial network states differed indicating a dependency on the past input $x_{\mathrm{stim}}$ (fig. 3D&E). Both the ubiquitous existence of memory, and the wide diversity of memory timescales were consistent with the observations made in the spiking models (figs. 1 and 2), suggesting that the degeneracy of weights was indeed the key factor in the memory by accident phenomenon. In fact, the only rules that did not yield memories were the purely non-Hebbian ones ($\theta_1 = 0$). We verified that these results held qualitatively when extending this toy model to higher dimensions (see Appendix and fig. S8).

Besides qualitatively reproducing the observation that most stabilizing plasticity rules exhibit memories, this toy model led us to formulate several predictions:

1. **Trade-off between memory and stability**: rules that elicit the strongest memories—longer-lasting and more robust— should be closest to unstable regions (fig. 3D&E).

2. **Learning rates and memory**: *a priori*, the role of the learning rate was unclear to us, as it influenced both the learning and forgetting phases. In the toy model, the learning rate of the rules did not affect RI—the fixed points reached—only the speed of convergence $\tau$. Therefore lower (non-zero) learning rates should lead to longer-lasting memories.

3. **Hebbian vs non-Hebbian terms and memory**: the model predicted that the only rules that do not elicit memory are exclusively non-Hebbian: $\frac{\partial w_i}{\partial t} \propto y - y*$. Further, rules combining the plasticity parameters could be more efficient, i.e. with longer-lasting memories, than exclusively Hebbian or non-Hebbian rules.

## 4.2 Feedforward implicit model

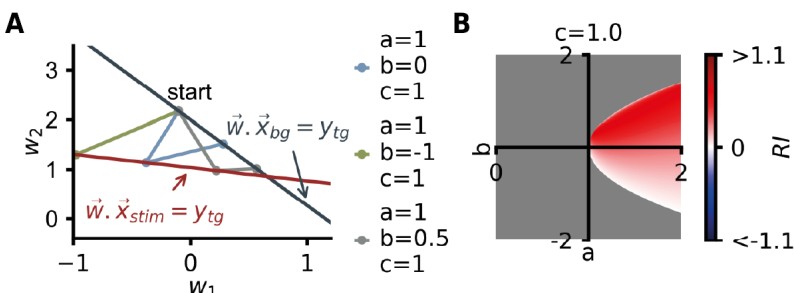

Figure 4: **A**: Fixed points for three rules of the implicit toy model in the A-B-A task, shown in weight space. All networks are initialized at the same state $w_0 \in W^*_{x_{bg}}$. **B**: Phase portrait of the relative improvement RI as a function of the values of $a$ and $b$, for fixed $c$. This was computed for $\Theta_{x_{bg}} = \frac{\pi}{6}$ and $\Theta_{x_{stim}} = \frac{\pi}{6} + \frac{\pi}{4}$. Grey denotes unstable rules.

One potential concern with our previous toy model (and indeed, all of our previous approaches) is the dependence on a particular parameterization of the learning rules. In this section we therefore defined a more abstract class of plasticity rules, operating in the same linear feedforward network as in the previous section. In the vein of recent work [28], we forego an explicit parameterization and instead considered rules that minimize different distance metrics in weight space. More specifically, we again considered learning rules that lead to a stable output firing of $y^*$ for *any* input $x$:

$$\forall (x, y_0, w_0) \in \mathbb{R}^{2 \times 1 \times 2}, \lim_{t \to +\infty} y(t, x, y_0, w_0) = y^* \tag{5}$$

Then we assumed that the stabilizing weight configuration that network would choose $W^*_x$ would be the closest according to some distance metric:

$$\forall (x, y_0, w_0) \in \mathbb{R}^{2 \times 1 \times 2}, \lim_{t \to +\infty} w(t, x, y_0, w_0) = \underset{w \in W^*_x}{\mathrm{argmin}} ||w - w_0||^2_\Sigma \tag{6}$$

with $||.||_\Sigma$ the norm induced by the Mahalanobis distance $D$:

$$\forall (w_1, w_2) \in \mathbb{R}^{2 \times 2}, D_\Sigma(w_1, w_2) = \sqrt{(w_1 - w_2)^T \Sigma^{-1} (w_1 - w_2)} \tag{7}$$

We defined $\Sigma^{-1} = \begin{pmatrix} a & b \\ b & c \end{pmatrix}$ leaving us with three plasticity parameters: $a, b$, and, $c$. To be a well-defined distance, $\Sigma^{-1}$ needs to be positive semidefinite leading to the further constraint that $a \geq 0$ and $ac - b^2 \geq 0$. From these assumptions, we could calculate the system's fixed point as a function of the initial state (see Supplementary for the full derivation). Despite only describing the fixed points, not the dynamics used to reach them, we found some matches between explicitly and implicitly parameterized rules (fig. S6). For instance, the rule minimizing the L2 norm ($a = 1, b = 0, c = 1$) corresponded to $\theta_0 = 0, \theta_1 = -1$ in the explicit model, which could also be seen as minimizing the L2 mean squared error between the current activity $y$ and the desired activity $y^*$ (see Supplementary).

Again, it appeared that with these stabilizing rules, memory by accident was the rule and not the exception (fig. 4B). This arose for almost all choices of metric minimization, independent of the exact

dynamics chosen to implement these weight updates, suggesting our ideas generalized beyond the particular plasticity rule parameterization we might choose.

## 5 Testing insights from toy models in spiking networks

Finally, we returned to the full recurrent spiking network with four co-active plasticity rules to test the predictions from our toy models.

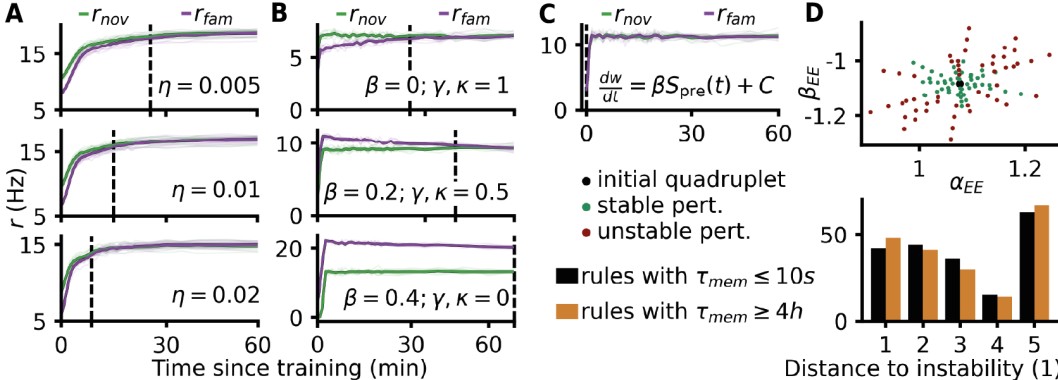

Figure 5: **A**: Recurrent spiking network with variants of iSTDP (I-to-E plasticity only) in the familiarity task, with different learning rates. Simulations were repeated 5 times, averages across seeds are shown in bold, dashed lines denote the last timestep at which the population firing rate in response to familiar stimuli was significantly different to novel responses (Student t-test, $p < 0.05$). **B**: Feedforward spiking network with I-to-E plasticity in the familiarity detection task. Three rules were tested from top to bottom: $\tau^{\text{pre}} = \tau^{\text{post}} = 20\text{ms}, \alpha = -0.12, \beta = 0, \kappa = \gamma = 1$ (original iSTDP) $\tau^{\text{pre}} = \tau^{\text{post}} = 20\text{ms}, \alpha = -0.12, \beta = 0.2, \kappa = \gamma = 0.5$ (half post-only, half Hebbian), $\tau^{\text{pre}} = \tau^{\text{post}} = 20\text{ms}, \alpha = -0.12, \beta = 0.4, \kappa = \gamma = 0$ (see fig. S7 for full simulations). **C**: Same as B, but for a different I-to-E plasticity rule with a spike-independent term, $\frac{dw}{dt} = \beta S_{\text{pre}}(t) + C$, with $\beta$ and $C$ plasticity parameters, and $S_{\text{pre}}$ the pre-synaptic spike train. **D**: Top: example base rule quadruplet in black, and stability of perturbations along 10 random directions in plasticity parameter space. Bottom: distribution of indices of the first unstable perturbation along each direction. Results for 10 quadruplets with long memory lifetimes $\tau_{\text{mem}} \geq 4\text{h}$ and 10 others with $\tau_{\text{mem}} \leq 10\text{s}$.

**Learning rate and memory**: We varied the learning rate in the recurrent network with only iSTDP (section 3.2); within the range tested, the slower the rule, the longer the memory lifetime (fig. 5A), as predicted.

**Trade-off between memory and stability**: we sought to verify that rule quadruplets with the longest memory lifetimes were those closest to the edge of the stability manifold. Calculating such distances involves simulating networks perturbed in all directions in the 24 dimensional parameter space, so is very compute-intensive. Here, therefore, we only report a trend when perturbing 10 stable rule quadruplets with memory lifetimes of 10s or less vs perturbing 10 stable rule quadruplets with memory lifetimes of 4h or more (fig. 5D). Overall, the distance to the instability border depended both on the rule quadruplet and on the direction chosen, and marginally more directions were immediately unstable for the long-memory quadruplets. Whether this is a reflection of a limited-compute budget or a real phenomenon remains to be seen.

**Hebbian vs non-Hebbian terms and memory**: we focused on I-to-E plasticity, to avoid potential confounds induced by co-active rules. Using mean-field analysis as in previous work [11], we derived the activity setpoint for spiking rules from eq. (1):

$$r_{\text{exc}} = \frac{-\alpha_{\text{IE}} r_{\text{inh}}}{\beta_{\text{IE}} + r_{\text{inh}}(\kappa_{\text{IE}} \tau_{\text{IE}}^{\text{post}} + \gamma_{\text{IE}} \tau_{\text{IE}}^{\text{pre}})} \qquad (8)$$

The iSTDP rule [11] chooses $\beta_{\text{IE}} = 0$, thus simplifying $r_{\text{inh}}$ out of the equation, which effectively ensures network stabilization for all inhibitory activity levels. In recurrent networks, rules with $\beta_{\text{IE}} \neq 0$ display unstable network dynamics (see fig. S4D). However, many unstable I-to-E rules were in fact stable as part of a quadruplet [20]. To be able to test these rules in a controlled setting,

we reverted to the feedforward spiking network. We tested I-to-E rules that had the same learning rates and target firing rates, but which traded off Hebbian terms $\kappa$ and $\gamma$ for the post-only term $\beta$. Surprisingly, the more post-only potentiation, the longer-lasting the memory (fig. 5B).

These experiments verified two predictions at once, though in an unexpected way. First, adding terms besides the Hebbian term could indeed be useful for memory-by-accident. Second, we found an unexpected verification of the stability-memory trade-off: the higher the $\beta$ which destabilizes the rule, the longer the memory. Finally, we verified that the only rule predicted not to elicit memory by accident, $\frac{\partial w_i}{\partial t} \propto y - y^*$, behaved as predicted in the spiking case. To do so, we defined an equivalent rule, $\frac{dw}{dt} = \beta S_{\text{pre}}(t) + C$. Note that such a rule was not part of the search space defined in eq. (1). We verified that it did not elicit memory by accident in the feedforward spiking model (fig. 5C).

## 6    Discussion

This study sought to understand a puzzling observation: almost all plasticity rules that produce stable network activity also lead to memory abilities [20]. By studying three different spiking network models we linked this phenomena to the degeneracy of weight matrices capable of stabilizing a network for a given input. In essence, from amongst the degenerate space of stable weights, the configuration that the plasticity rules chose depends on past inputs, creating a form of memory. We demonstrated the existence of this memory-by-accident phenomena all the way down to a 3-neuron linear rate model. We then used these tractable toy models to understand the phenomenon's ubiquity, finding it in nearly all rules, whether parameterized explicitly or implicitly. Instead of a seemingly fortunate accident, our analysis leads us to pose basic memory abilities as a near-unavoidable consequence of network stabilization.

In our toy models the only non-memorizing rules were purely non-Hebbian; do our findings boil down to "Hebbian learning creates memories"? We think not. While the link between Hebbian rules and memory is natural and longstanding, we find a wide diversity of combinations of Hebbian and non-Hebbian terms that produce long-lasting memories. Further, we found in our feedforward spiking models that the rules with the longest-lived "memories-from-accidents" were exclusively non-Hebbian (fig. 5B). Rather, we argue this is a generic property of stabilizing plasticity rules.

A promising aspect of memory by accident is that it seems to benefit from the system's complexity: the more neurons and weights, the more degenerate solutions; the more co-active learning rules, the more opportunities to exploit this degeneracy. This near-trivial phenomenon in toy models transferred to large recurrent network models and elicited complex and long-lasting memory abilities. Our recurrent spiking models, though relatively complex by today's standards, are but a simplistic reflection of brain regions, which are composed of orders of magnitude more neurons and synapses, and use hundreds of different synapse types [29], each of which could have their own plasticity rule. For now we can only speculate on the potential computations that this phenomenon could unlock in systems of comparable scale to the brain.

The memories we have been discussing exhibit a diversity of timescales, from seconds to hours (and potentially beyond). While our longer-lasting memories look like classic episodic memories that have often been modeled with spike-timing dependent plasticity rules like ours, the shorter of these timescales instead match those discussed by the activity-silent working memory literature [25]. Interestingly, classic models in this area rely on qualitatively different plasticity mechanisms such as short-term plasticity [30, 31]. It is exciting that our theory proposes a unifying explanation for both phenomena operating on different timescales.

The systematic emergence of basic forms of memory from relatively simple unsupervised and local plasticity rules could also provide hints on the evolution of the complex learning abilities observed. For plastic neural networks to be useful they must be stable. We find that as soon as they are stable they permit memory, potentially presenting an easy stepping stone to higher-order cognitive abilities from compositions of readily available memory motifs.

**Limitations**: Given the compute-load of simulating large plastic recurrent spiking networks with a high dimensional plasticity space, we could only partially verify the predictions made by the toy models. Moreover, many other phenomena influence the memory of the system besides those captured in the toy model. For instance, we noticed that some rule quadruplets don't respond to any

memory task, but have very high firing rates ($> 30$Hz), effectively making them recurrent driven and oblivious to their inputs.

We have also not modeled the effects of co-active rules, which in practice appeared to be a key element to make memories generated by the pairwise rules robust and noticeable on behavioural timescales [20]. Indeed, most of the original set of co-active plasticity rules used as a starting point for this study [20] were unstable when considered in isolation, or in a mean-field model with co-activity (fig. S1). This effectively prevented us from drawing conclusions on their memory capacity in this work. However, understanding how rules unstable in isolation can be stable together and produce longer-lasting memories than their individual counterparts is an important avenue for future work, perhaps using more refined versions of mean-field analysis than was performed here [32, 9, 10, 33, 7].

We might also wonder how these mechanisms could be extended to embed lifelong memories. In our networks it appears that memories are ultimately erased by random fluctuations (fig. 2C), something not captured by our toy models. However, as a first approximation, the larger the RI (the further away the final state is from the initial), the longer the memory will last before being erased by noise. Further, some learning rules seem to be able to dramatically postpone this eventual forgetting, especially through co-activity (fig. 2A). Together these mechanisms seem sufficient to encode a memory long enough for it to be consolidated by other systems, such as the wake-sleep cycle. On the contrary, a memory system with graceful forgetting and robust stability enforced as default may be desirable.

To sum up, our distillation of automatically discovered plasticity rules in large recurrent networks resulted in a remarkably general and simple phenomenon, memory by accident, that highlights the unreasonable effectiveness of simple unsupervised rules at making memories.

## Acknowledgments and Disclosure of Funding

We thank Tim Vogels for his help and support, as well as Lukas Braun, Everton Agnes, Peter Latham and Antonio Sclocchi for useful discussions. This work was supported by a Schmidt Science Polymath Award, the Sainsbury Wellcome Centre Core Grant from Wellcome (219627/Z/19/Z) and the Gatsby Charitable Foundation (GAT3850).

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

# A Technical Appendices and Supplementary Material

The code and data to reproduce the results in this paper can be found on Github. Spiking network simulations were performed on a single CPU using the Auryn simulator [34]. The perturbations of rule quadruplets in fig. 5C ($\approx$3000 simulations) were performed on 500 CPUs on the ISTA HPC cluster. Rate models and analysis of all simulations was performed using numpy [35], JAX [36], matplotlib [37], SciPy [38] and scikit-learn [39].

## A.1 Recurrent spiking network model

### A.1.1 Network model

We considered a recurrent spiking network with $N_\mathrm{E} = 4096$ excitatory neurons and $N_\mathrm{I} = 1024$ inhibitory neurons (leaky-integrate and fire point neurons with variable threshold, AMPA and NMDA currents, and conductance-based synapses). This network was based on previous work [8, 20]. The membrane potential dynamics of neuron $j$ (excitatory or inhibitory) followed:

$$\tau_m \frac{\mathrm{d}}{\mathrm{d}t} V_j(t) = -\left(V_j(t) - V_\mathrm{rest}\right) - g_j^\mathrm{E}(t)\left(V_j(t) - E_\mathrm{E}\right) - g_j^\mathrm{I}(t)\left(V_j(t) - E_\mathrm{I}\right), \tag{9}$$

with $\tau_m = 20$ ms, $V_\mathrm{rest} = -70$ mV, $E_\mathrm{E} = 0$ mV and $E_\mathrm{I} = -80$ mV.

A postsynaptic spike occurred whenever the membrane potential $V_j(t)$ crossed a threshold $V_j^\mathrm{th}(t)$, with an instantaneous reset to $V_\mathrm{reset} = -70$ mV. This threshold $V_j^\mathrm{th}(t)$ was incremented by $V_\mathrm{spike}^\mathrm{th} = 100$ mV every time neuron $j$ spiked and otherwise decayed following:

$$\tau_\mathrm{th} \frac{\mathrm{d}}{\mathrm{d}t} V_j^\mathrm{th}(t) = V_\mathrm{base}^\mathrm{th} - V_j^\mathrm{th}(t), \tag{10}$$

with $V_\mathrm{base}^\mathrm{th} = -50$ mV. The excitatory and inhibitory conductances, $g^\mathrm{E}$ and $g^\mathrm{I}$ evolved such that

$$g_j^\mathrm{E}(t) = a g_j^\mathrm{AMPA}(t) + (1 - a) g_j^\mathrm{NMDA}(t) \quad \text{and}$$

$$\frac{\mathrm{d}}{\mathrm{d}t} g_j^\mathrm{I}(t) = -\frac{g_j^\mathrm{I}(t)}{\tau_\mathrm{GABA}} + \sum_{i \in \mathrm{Inh}} w_{ij}(t) S_i(t)$$

$$\text{with} \quad \frac{\mathrm{d}}{\mathrm{d}t} g_j^\mathrm{AMPA}(t) = -\frac{g_j^\mathrm{AMPA}(t)}{\tau_\mathrm{AMPA}} + \sum_{i \in \mathrm{Exc}} w_{ij}(t) S_i(t) \quad \text{and} \tag{11}$$

$$\frac{\mathrm{d}}{\mathrm{d}t} g_j^\mathrm{NMDA}(t) = \frac{g_j^\mathrm{AMPA}(t) - g_j^\mathrm{NMDA}(t)}{\tau_\mathrm{NMDA}},$$

with $w_{ij}(t)$ the connection strength between neurons $i$ and $j$ (unitless), $a = 0.23$ (unitless), $\tau_\mathrm{GABA} = 10$ ms, $\tau_\mathrm{AMPA} = 5$ ms, $\tau_\mathrm{NMDA} = 100$ ms, $S_i(t) = \sum \delta(t - t_i^*)$ the spike train of presynaptic neuron $i$, where $t_i^*$ denotes the spike times of neuron $k$, and $\delta$ the Dirac delta.

The network was initialized with random sparse connectivity (10%), with $w_\mathrm{EE}^\mathrm{init} = w_\mathrm{EI}^\mathrm{init} = 0.1$ and $w_\mathrm{IE}^\mathrm{init} = w_\mathrm{II}^\mathrm{init} = 1$.

The excitatory neurons in the network received $N_{\mathrm{inp} \to \mathrm{E}} = 11025$ inputs from Poisson neurons firing at $r_\mathrm{bg}^\mathrm{inp} = 10 Hz$. When a stimulus was active, a subset of the input neurons increased their firing rate to $r_\mathrm{active}^\mathrm{seq} = 100 Hz$. The connectivity from input neurons to excitatory and inhibitory neurons was receptive-field-like: for each recurrent neuron, we selected a random input neuron as the center of the circular receptive field of radius 8. The connections from neurons of this circular patch of input neurons to the considered recurrent neuron was $w_\mathrm{inp} = 0.075$, and 0 to all other input neurons. The inhibitory neurons received inputs from $N_{\mathrm{inp} \to \mathrm{I}} = 4096$ Poisson neurons with $w_\mathrm{inp}$ and similar receptive field connectivity than for the excitatory population. However, inhibitory neurons only received background inputs ($r_\mathrm{bg}^\mathrm{input}$) and no specific stimulus patterns.

### A.1.2 Plasticity parameterization

This parameterization of plasticity rules included variations of spike-timing-dependent plasticity, and was taken from previous work [19, 20]. The weight from neuron $i$ to neuron $j$ of type $X$ and $Y$ (excitatory or inhibitory) evolved such that:

$$\frac{\mathrm{d}w_{ij}(t)}{\mathrm{d}t} = \eta[S_i(t)\left(\alpha_{\mathrm{XY}} + \kappa_{\mathrm{XY}}x_j(t)\right) + S_j(t)\left(\beta_{\mathrm{XY}} + \gamma_{\mathrm{XY}}x_i(t)\right)] \tag{12}$$

with $\eta = 0.01$ a fixed learning rate, $S_i(t) = \sum_k \delta(t - t_k^i)$ the spike train of neuron $i$, $\delta$ the Dirac delta function to denote the presence of a pre (post)-synaptic spike at time $t$. The synaptic traces $x_i$ and $x_j$ are low-pass filters of the activity of presynaptic neuron $i$ and postsynaptic neuron $j$, with time constants $\tau_{\mathrm{pre}}$ and $\tau_{\mathrm{post}}$, such that:

$$\frac{\mathrm{d}}{\mathrm{d}t}x_i(t) = -\frac{x_i(t)}{\tau_{\mathrm{pre}}^{\mathrm{XY}}} + S_i(t) \quad \text{and} \quad \frac{\mathrm{d}}{\mathrm{d}t}x_j(t) = -\frac{x_j(t)}{\tau_{\mathrm{post}}^{\mathrm{XY}}} + S_j(t), \tag{13}$$

Overall, this search space comprised 6 tunable plasticity parameters per synapse type XY (X, Y $\in$ (E, I)): $\theta_{\mathrm{XY}} = [\alpha_{\mathrm{XY}}, \beta_{\mathrm{XY}}, \gamma_{\mathrm{XY}}, \kappa_{\mathrm{XY}}, \tau_{\mathrm{pre}}^{\mathrm{XY}}, \tau_{\mathrm{post}}^{\mathrm{XY}}]$, for a total of 24 plasticity parameters across all four synapse types.

Note that all weights in the network were capped at all times, in the $[0, w_{\mathrm{max}}]$ range, with $w_{\mathrm{max}} = 20$, though rule quadruplets in [20] were considered unstable if more than 10% of the weights at any synapse type reached these extreme values.

### A.1.3 Familiarity detection task

We considered a familiarity detection task that was similar to previous work [20]. The network first received nonspecific background inputs for 1h—pre-training phase—, followed by a 40s training phase during which four non-overlapping input patterns—the familiar stimuli—were active in alternation. When a given stimulus was active, $\approx$10% of the input neurons to the excitatory population had elevated firing rates, while the others remained at background. After training, we reverted to background inputs and regularly probed the network with familiar and novel stimuli for an hour—post-training phase.

Given the input structure and connectivity described above, each stimulus, novel or familiar excited a different subset of recurrent neurons. We defined "engrams" for each stimulus pattern (novel or familiar) by probing the network before training started, and labeling the top 10% of excitatory and inhibitory neurons as part of the engram for the presented stimulus. In practice, since the stimuli were non overlapping, the engrams defined this way also had little overlap.

Note that each stimulus elicited a different network response, in the sense that each stimulus preferentially excited a different subset of recurrent neurons (this can be seen in fig. S2) and thus the stimulus identity could be decoded from the population vector of neuron activities. However, whether this stimulus has been encountered by the network in the past —its novelty or familiarity— was unknown.

We chose a simple decoding strategy for stimulus familiarity: the mean firing rate of the excitatory population (each excitatory neuron contributes equally to the decoding). In the paper that inspired this work [20], a Student t-test was performed over several seeds/simulations/stimuli over mean firing rates in response to familiar or novel stimuli to determine whether novel stimuli elicited statistically different mean firing rates than familiar stimuli. Note that by design, all stimuli elicited statistically indistinguishable mean firing rates in static or naive plastic networks. Thus this task flagged a stimulus-specific change in network activity due to synaptic plasticity.

## A.2 Feedforward spiking network model

### A.2.1 Network model

1000 Poisson neurons (800 excitatory and 200 inhibitory) projected onto a single output neuron, with the same neuron model and parameters as in the previous section. The excitatory weights were fixed at $w_{ee} = 0.1$, the inhibitory weights were plastic with the parameterization defined for the recurrent spiking case, and initialized at $w_{ie} = 1$.

Note that in fig. 5C, we used a different learning rule, not part of the plasticity parameterization described above. For this rule, $\frac{dw}{dt} = \beta S_{\mathrm{pre}}(t) + C$ with $\beta = 0.4$ and $C = 0.00012$ to obtain a target firing rate of 3Hz given the integration time-step of the simulation (0.1ms).

### A.2.2 Familiarity detection task

The task closely resembled the task in the recurrent case. During a pre-training phase of 1h, all input neurons fired at 10Hz. During the training phase that lasted 60s, 100 excitatory and 25 inhibitory increased their firing rates to 100Hz and 50Hz respectively (the "familiar" stimulus). After the training phase, the network was regularly probed on its network response to the familiar stimulus and another, novel stimulus of the same structure than the familiar stimulus but with different neurons (no overlap).

## A.3 Toy model 1: Explicit parameterization

### A.3.1 2D version

The model is a linear feedforward network with two inputs $\boldsymbol{x} = (x_0, x_1)$ projecting on a single output neuron $y$:

$$y(t) = w_0(t)x_0(t) + w_1(t)x_1(t) \tag{14}$$

Besides, $\forall t \geq 0,\ y(t) \geq 0,\ x_0(t) \geq 0,\ x_1(t) \geq 0;\ ||\boldsymbol{x}|| = 1$ with $||.||$ the L2 norm. Weights were plastic and unconstrained.

Initially, we considered a four-parameter set of Hebbian/non-Hebbian plasticity rules inspired by the full spiking model (see mean-field section below for the relationship between spike-based and rate-based plasticity):

$$\frac{\partial w_i(t)}{\partial t} = \eta\big(\theta_0 + \theta_1 x_i(t) + \theta_2 y(t) + \theta_3 x_i(t)y(t)\big) \tag{15}$$

with $\theta_0$, $\theta_1$, $\theta_2$ and $\theta_3$ four plasticity parameters, and $\eta = 0.01$ a fixed learning rate (omitted below). From this full search space, we only considered rules that admitted a target output firing rate $y^* \in \mathbb{R}^+$ as a stable fixed point for all inputs $\boldsymbol{x}$ considered here. The existence of $y^*$ as a fixed point implied that for all inputs $x_i$

$$\theta_0 + \theta_1 x_i + \theta_2 y^* + \theta_3 x_i y^* = 0 \implies \theta_0 = -\theta_2 y^* \text{ and } \theta_1 = -\theta_3 y^* \tag{16}$$

Thus the search space became two-dimensional:

$$\begin{cases} \dfrac{\partial w_0}{\partial t} = (y - y^*)(\theta_0 + \theta_1 x_0) \\ \dfrac{\partial w_1}{\partial t} = (y - y^*)(\theta_0 + \theta_1 x_1) \end{cases} \implies \dot{\mathbf{w}} = A\mathbf{w} + B \tag{17}$$

with $A = \begin{pmatrix} x_0(\theta_0 + \theta_1 x_0) & x_1(\theta_0 + \theta_1 x_0) \\ x_0(\theta_0 + \theta_1 x_1) & x_1(\theta_0 + \theta_1 x_1) \end{pmatrix}$ and $B = -y^* \begin{pmatrix} \theta_0 + \theta_1 x_0 \\ \theta_0 + \theta_1 x_1 \end{pmatrix}$.

This system has only one non-zero eigenvalue: $\lambda_1 = \theta_0(x_0 + x_1) + \theta_1(x_0^2 + x_1^2)$. The system thus has a neutral mode ($\lambda_0 = 0$), for the system to converge to a point on the line attractor, we need $\lambda_1 < 0$. Because $||\mathbf{x}|| = 1$ and $x_0, x_1 \geq 0$, $x_0 + x_1 \in [1, \sqrt{2}]$ and $x_0^2 + x_1^2 = 1$. As a result, we need $\theta_0$ and $\theta_1$ to be below the lines $\theta_0 + \theta_1 = 0$ and $\theta_0\sqrt{2} + \theta_1 = 0$.

For the numerical results reported in the paper, we simulated the system in the A-B-A task with: $y^* = 1$, $\boldsymbol{w}_0 = (1, 0.27) \in \boldsymbol{W}^*_{\boldsymbol{x}_{\mathrm{bg}}}$, $\boldsymbol{x}_{\mathrm{bg}} \angle \boldsymbol{x}_{\mathrm{stim}} = \frac{\pi}{4}$. We ran the system until convergence at each phase of the A-B-A task, in practice we found that $T = 20000$ epochs was sufficient.

We verified that the results of the parameter sweeps on $\theta_0$ and $\theta_1$ had similar trends for different stimuli angles and initializations.

### A.3.2 Extension to higher input dimensions

In the main text, we chose the smallest model possible (2D) for ease of visualization. However, the findings readily extend to higher dimensions ($N_{\text{in}} > 2$ input neurons).

We consider $N_{\text{in}}$ input neurons with activity $\boldsymbol{x}$ projecting on a single output neuron $y$ with weights $\boldsymbol{w}$: $y = \boldsymbol{W}^T \boldsymbol{x}$. We choose $\boldsymbol{x}$ to be of unit norm with nonnegative entries. The plasticity rules are the same as in the 2D case:

$$\frac{\partial w_i}{\partial t}(t) = (y(t) - y^*)\big(\theta_0 + \theta_1 x_i(t)\big), \ i = 1, \cdots, N_{\text{in}} \tag{18}$$

This is an affine system of ODEs, which can be written in vector form as $\dot{\boldsymbol{w}} = A\boldsymbol{w} + \boldsymbol{b}$, with $A = \boldsymbol{x}(\theta_0 \mathbf{1} + \theta_1 \boldsymbol{x})$, and $\mathbf{1}$ a $N_{\text{in}}$-dimensional vector of ones.

For a constant input $\mathbf{x}$, this system is rank one, and the non-zero eigenvalue is $\lambda = \theta_0 \sum_{i=1}^{N_{\text{in}}} x_i + \theta_1 \sum_{i=1}^{N_{\text{in}}} x_i^2 = \theta_0 \sum_{i=1}^{N_{\text{in}}} x_i + \theta_1$ for unitary norm inputs. Note that this is a generalization of the derivation presented above.

Edge of stability: For the system to be stable, we need $\lambda < 0$, which translate for unit-norm, nonnegative inputs to $\theta\sqrt{N_{\text{in}}} + \theta_1 < 0$ and $\theta_0 + \theta_1 < 0$.

Defining a metric to evaluate memory: As in the 2D case, we define $\boldsymbol{w}_{\text{bg}}$, the steady state weights for input $\boldsymbol{x}_{\text{bg}}$ at the start of the A-B-A task, and $\boldsymbol{w}_{\text{bg}'}$ are those for $\boldsymbol{x}_{\text{bg}}$ at the end of the task. However, the 2D definition of $RI$ (eq. (4)) does not generalize readily to N-dimensions, as $\boldsymbol{w}_{\text{bg}\cap\text{stim}}$, the intersection between the hyperplanes $\boldsymbol{W}^*_{\boldsymbol{x}_{\text{bg}}} : \boldsymbol{w}^T \boldsymbol{x}_{\text{bg}} = y^*$ and $\boldsymbol{W}^*_{\boldsymbol{x}_{\text{stim}}} : \boldsymbol{w}^T \boldsymbol{x}_{\text{stim}} = y^*$ is not unique (assuming these hyperplanes are not parallel). As a proxy, we define $RI = ||\boldsymbol{w}_{\text{bg}'} - \boldsymbol{w}_{\text{bg}}||$, which only evaluates how far the final state is from the initial one, and not whether the change is pushing towards the intersection or not.

Overall, as can be seen in fig. S8 increasing the dimensionality of the toy models did not change qualitatively the findings reported in the main paper.

### A.4 Toy model 2: Implicit parameterization

#### A.4.1 2D version

This toy model only described the fixed point reached by an implicitly-defined class of learning rules operating in the same linear feedforward network as in previous section.

Specifically, we made two assumptions on the learning rules:

$$\text{Stabilization:} \forall (\boldsymbol{x}, y_0, \boldsymbol{w}_0) \in \mathbb{R}^{2 \times 1 \times 2}, \lim_{t \to +\infty} y(t, \boldsymbol{x}, y_0, \boldsymbol{w}_0) = y^* \tag{19}$$

$$\text{Distance minimization:} \forall (\boldsymbol{x}, y_0, \boldsymbol{w}_0) \in \mathbb{R}^{2 \times 1 \times 2}, \lim_{t \to +\infty} \boldsymbol{w}(t, \boldsymbol{x}, y_0, \boldsymbol{w}_0) = \underset{\boldsymbol{w} \in \boldsymbol{W}^*_{\boldsymbol{x}}}{\text{argmin}} ||\boldsymbol{w} - \boldsymbol{w}_0||^2_{\Sigma} \tag{20}$$

with $||.||_{\Sigma}$ the norm induced by the Mahalanobis distance $D$:

$$\forall (\mathbf{w}_1, \mathbf{w}_2) \in \mathbb{R}^{2 \times 2}, \ D_{\Sigma}(\mathbf{w}_1, \mathbf{w}_2) = \sqrt{(\mathbf{w}_1 - \mathbf{w}_2)^T \Sigma^{-1}(\mathbf{w}_1 - \mathbf{w}_2)} \tag{21}$$

with $\Sigma^{-1} = \begin{pmatrix} a & b \\ b & c \end{pmatrix}$, where $a, b,$ and $c$ are the three plasticity parameters in this search space. To be a well-defined distance, $\Sigma^{-1}$ needs to be positive semi-definite leading to the further constraint that $a \geq 0$ and $ac - b^2 \geq 0$.

From these assumptions, we derived the final network state $\boldsymbol{w}^f = (w_0^f, w_1^f)$ as a function of the initialization $\boldsymbol{w}^i = (w_0^i, w_1^i)$, $\boldsymbol{x}^i$ (input for which the network is initialized), and (fixed) input $\boldsymbol{x}^f$. Since the final state belongs to $\boldsymbol{W}^*_{\boldsymbol{x}^f}$, we have $\mathbf{w}^{f^T}\mathbf{x}^f = y^* \implies w_1^f = \frac{y^* - w_0^f x_0^f}{x_1^f}$. We minimize the distance $D$:

$$D(\mathbf{w}, \mathbf{w}^i)^2 = \frac{||\mathbf{x}^f||_{\Sigma^{-1}}^2}{x_1^{f\,2}} w_0^2 + 2 \frac{-a w_0^i x_1^{f\,2} + b x_1^f \left[ y^* + w_0^i x_0^f - w_1^i x_1^f \right] + c x_0^f \left[ w_1^i x_1^f - y^* \right]}{x_1^{f\,2}} w_0$$

(22)

$$+ ||\mathbf{w}^i||_{\Sigma}^2 + \frac{y^* \left[ -2 b w_0^i x_1^f + c(-2 w_1^i x_1^f + y^*) \right]}{x_1^{f\,2}}$$

(23)

Since $\frac{||\mathbf{x}^f||_{\Sigma^{-1}}^2}{x_1^{f\,2}} > 0$, the expression above has a single minimum:

$$w_0^f = \frac{a w_0^i x_1^{f\,2} + b x_1^f (w_1^i x_1^f - w_0^i x_0^f - y^*) + c x_0^f (y^* - w_1^i x_1^i)}{||\mathbf{x}^f||_{\Sigma^{-1}}^2}$$

(24)

We applied the result above twice, for each phase of the A-B-A task.

**Relationship between explicit and implicit toy models**: although we don't have a general mapping from the implicit to the explicit rules in both toy models, some rules have the same steady states in both cases.

Notably, the Hebbian rule $\frac{\partial w_i}{\partial t} = -x_i(y - y^*)$ ($\theta_0 = 0, \theta_1 = -1$) had identical fixed point to the rule minimizing the Euclidean distance in the implicit model ($a = 1, b = 0, c = 1$), see fig. S6D. The explicit form of this rule could also be seen as the gradient descent update wrt the loss function $\mathcal{L} = \frac{(y - y^*)^2}{2}$, i.e. $\frac{\partial \mathcal{L}}{\partial w_i} = x_i(y - y^*) = -\frac{\partial w_i}{\partial t}$.

### A.4.2 Extension to higher input dimensions

This toy model has the same network architecture as above, but the number of input neurons does change the number of plasticity parameters, as the rules are this time parameterized by a distance metric of the Mahalanobis family (with covariance $\Sigma \in \mathbb{R}^{N_{\text{in}} \times N_{\text{in}}}$, leaving us with $\frac{N_{\text{in}}(N_{\text{in}}+1)}{2}$ plasticity parameters.

Edge of stability: This corresponds to $\Sigma$ losing its positive semi-definite property (i.e. at least one eigenvalue becomes 0).

### A.5 Toy-model 3: Mean-field-inspired model

A common method to study spike-timing-dependent plasticity is to perform mean-field analysis [33, 6, 11], which assumes a large network of uncorrelated neurons. Under these assumptions, the weight updates in the spiking network eq. (1) become:

$$\langle \frac{\mathrm{d}w(t)}{\mathrm{d}t} \rangle = \eta [ r_{pre} \alpha_{XY} + r_{post} \beta_{XY} + r_{pre} r_{post} (\kappa_{XY} \tau_{XY}^{post} + \gamma_{XY} \tau_{XY}^{pre}) ]$$

(25)

with $r_{pre}$ and $r_{post}$ the firing rates of the pre- and post-synaptic neurons. This method allowed us to get a rate "equivalent" of each spike-timing dependent rule defined in eq. (1). We embedded the rate-equivalent rule quadruplets in a 2-neuron linear recurrent network (2RNN) undergoing the familiarity detection task (fig. S1). The activities $r_E, r_I$ of the 2RNN, representing the excitatory and inhibitory spiking populations, followed:

$$\begin{cases} r_E(t+1) = w_{ee}(t) r_E(t) - w_{ie}(t) r_I(t) + x_E(t) \\ r_I(t+1) = w_{ei}(t) r_E(t) - w_{ii}(t) r_I(t) + x_I(t) \end{cases}.$$

(26)

The activities $r$ and weights $w$ were constrained to be positive at all times. The familiarity task was ported to this setting by providing background input to the network $\mathbf{x}_{bg} = (1, 1)$, followed by a training period with stimulus $\mathbf{x}_{stim} = (1.5, 1)$ before reverting to $\mathbf{x}_{bg}$. Over 95% of the rule quadruplets that were stable in the recurrent spiking model elicited diverging weight or activity dynamics in the rate model, such as the rule quadruplet shown in fig. 1 (fig. S1B). Nevertheless, this 2RNN satisfyingly approximated some rules, particularly those evolving in isolation, such as the rate-equivalent of iSTDP (fig. S1C).

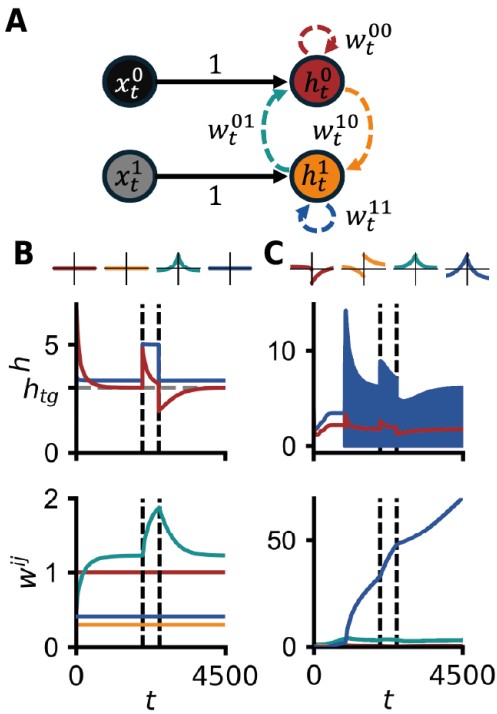

Supplementary Figure S1: **Mean-field-inspired model.** **A**: Linear 2-neuron recurrent network (2RNN) and notations. **B**: 2RNN evolving with the rate-equivalent of the iSTDP rule during the familiarity task (see fig. 2A for spiking equivalent). Dashed lines denote the onset and offset of training. Top: network activities during the task. Bottom: evolution of the four recurrent weights. Colors match the cartoon in A. **C**: Same as B, but for the meta-learned rule quadruplet shown in fig. 1.

Overall this suggested that the assumptions made to obtain the rate equivalent rules were not valid in the case of co-active rules. Indeed, the one rule for which the 2RNN model performed qualitatively similarly like the spiking model is iSTDP, which was shown to decorrelate neuronal activities [11], thus ensuring that the assumption of uncorrelated neuron activities holds. We thus moved to a more abstract setting to understand the memory by accident phenomenon.

## A.6 Supplementary figures

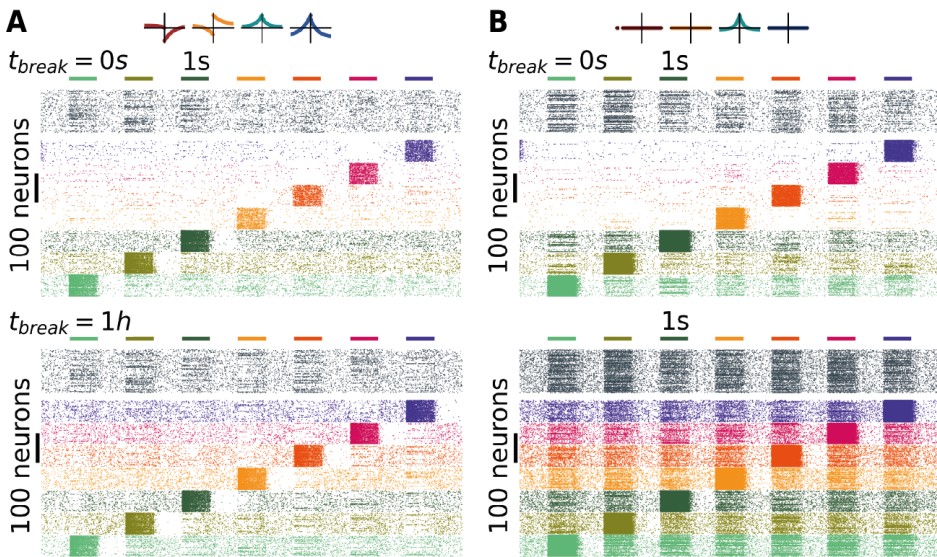

Supplementary Figure S2: **Visualization of network activities for fig. 1. A,B**: Raster plots during two test sessions of the familiarity detection task. Neurons are colored by which engram they belong to (see methods for definition of engrams, gray shows neurons not part of any engram).

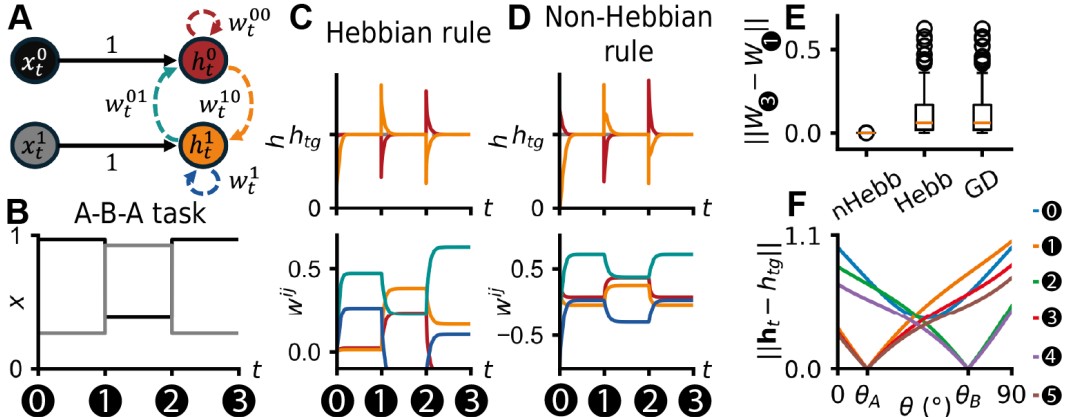

Supplementary Figure S3: **A linear RNN reproduces aspects of memory by accident A**: Network and notations, activities are restricted to be positive. **B**: Example input activities in the A-B-A task. Each stimulus presentation is chosen to be long enough for any potential fixed point to be reached. **C**: Hebbian rule in the A-B-A task: $\Delta w_t^{pre\,post} \propto (h_{tg} - h_t^{post})h_t^{pre}$ with $h_{tg}$ a fixed target (1). Top: Recurrent neurons activities, Bottom: 4 network weights. **D**: Same as C for a non-Hebbian rule: $\Delta w_t^{pre\,post} \propto (h_{tg} - h_t^{post})$. **E**: Distance between the network weights at the end of the first or the second presentation of stimulus A: averaged over many simulations for the three learning rules tested. "GD" is online gradient descent on the mean squared error loss of the network activity compared to the target activity $h_{tg}$. **F**: Network response profile for stimuli of various angles at different timepoints of the A-B-A task.

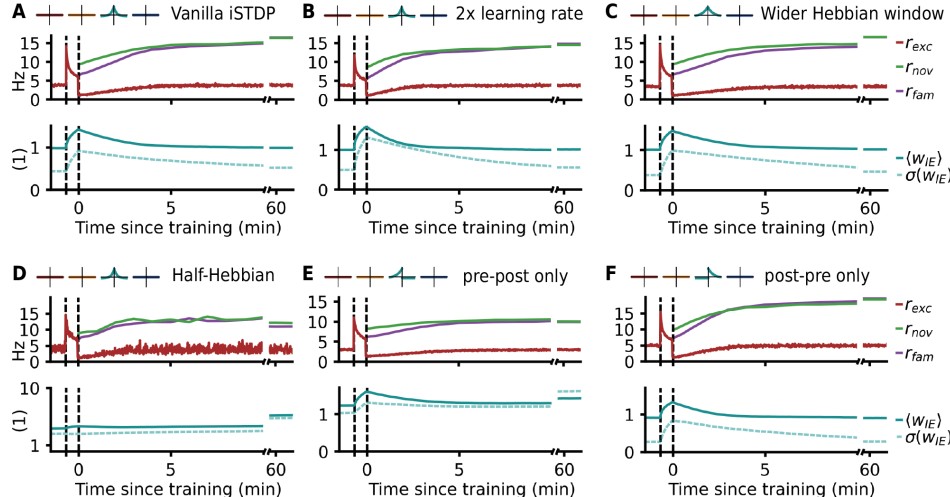

Supplementary Figure S4: **Variations of iSTDP and memory by accident**: Recurrent spiking network undergoing the familiarity detection task, for 6 variants of the iSTDP rule. All variants have the same target excitatory rate of 3Hz. **A**: $\tau_{IE}^{pre} = \tau_{IE}^{post} = 20ms, \alpha_{IE} = -0.12, \beta_{IE} = 0, \kappa_{IE} = \gamma_{IE} = 1$. **B**: $\tau_{IE}^{pre} = \tau_{IE}^{post} = 20ms, \alpha_{IE} = -0.24, \beta_{IE} = 0, \kappa_{IE} = \gamma_{IE} = 2$. **C**: $\tau_{IE}^{pre} = \tau_{IE}^{post} = 40ms, \alpha_{IE} = -0.12, \beta_{IE} = 0, \kappa_{IE} = \gamma_{IE} = 0.5$. **D**: $\tau_{IE}^{pre} = \tau_{IE}^{post} = 20ms, \alpha_{IE} = -0.12, \beta_{IE} = 0.06, \kappa_{IE} = \gamma_{IE} = 0.5$. **E**: $\tau_{IE}^{pre} = \tau_{IE}^{post} = 20ms, \alpha_{IE} = -0.12, \beta_{IE} = 0, \kappa_{IE} = 0, \gamma_{IE} = 1$. **F**: $\tau_{IE}^{pre} = \tau_{IE}^{post} = 20ms, \alpha_{IE} = -0.12, \beta_{IE} = 0, \kappa_{IE} = 1, \gamma_{IE} = 0$.

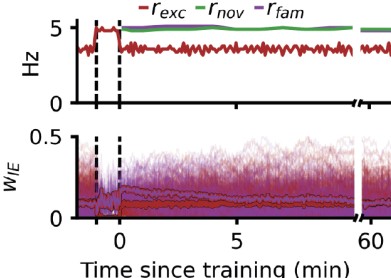

Supplementary Figure S5: **Feedforward spiking mode with E-to-E plasticity.** Top: output neuron firing rate. Bottom: evolution of the 800 excitatory (plastic) weights. Weights from all excitatory input neurons are in red, the subset of weights belonging to the familiar stimulus are overlaid in purple. The means of the two groups ("familiar" weights vs rest) are in bold. The plasticity rule used is $\tau_{EE}^{pre} = \tau_{IE}^{post} = 100ms, \alpha_{EE} = 1, \beta_{EE} = -0.8, \kappa_{EE} = \gamma_{EE} = -1$.

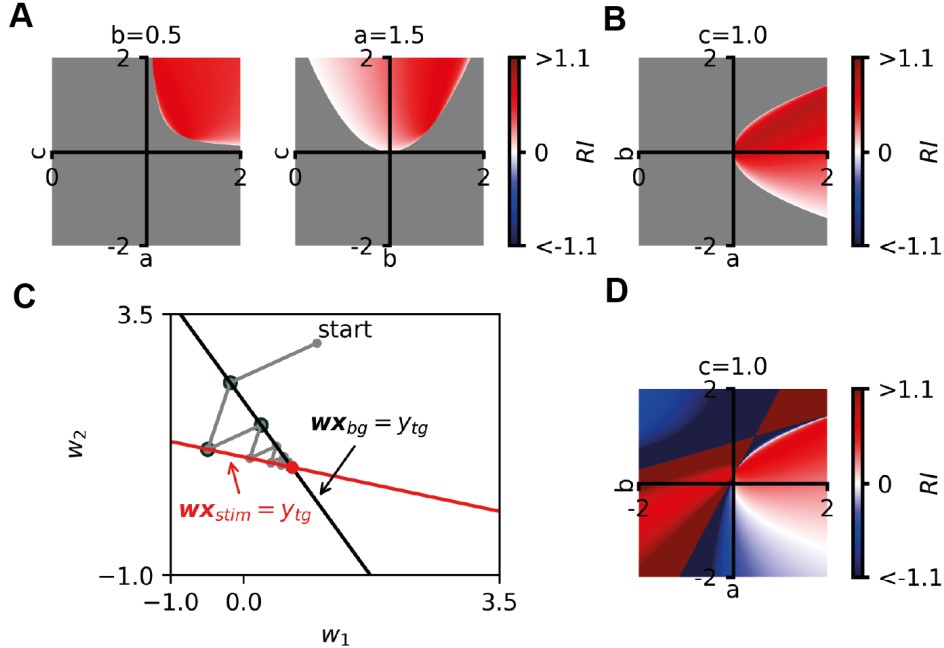

Supplementary Figure S6: **Additional analysis of the implicit feedforward toy model**: **A**: Similar parameter sweeps as in fig. 4B, but varying other parameter combinations. **B**: Same parameter sweep as in fig. 4B, but for an angle of $\frac{\pi}{3}$ between the two inputs. **C**: Grey: dynamics of the $\theta_0 = 0, \theta_1 = -1$ rule from the explicit parameterization in the A-B-A task. Black dots represent the fixed points of the rule associated to $a = 1, b = 0, c = 0$ in the implicit parameterization. **D**: Parameter sweep on the plasticity rules (varying $a$ and $b$, $c$ fixed. But relaxing the assumption that $\Sigma^{-1}$ needs to be positive semi-definite.

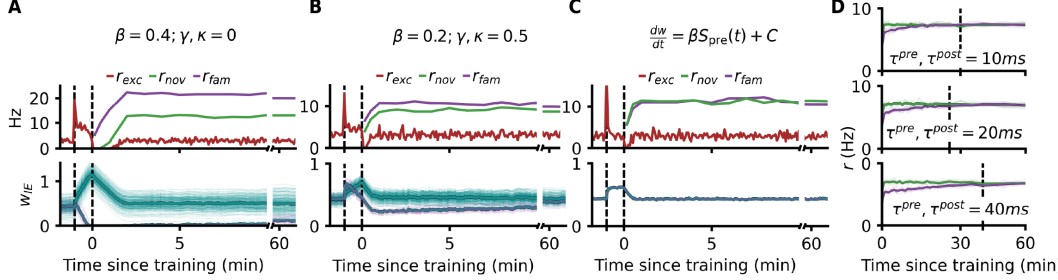

Supplementary Figure S7: **Additional analysis on testing predictions from toy models. A, B, C**: Top: firing rate of the post-synaptic neuron during simulation associated to fig. 5B&C (red), as well as the firing rate in response to the novel and familiar stimuli. Bottom: inhibitory weights, weights from inhibitory input neurons aprt of the familiar stimulus are in purple, the rest is in teal. Averages of the two groups are in bold. Dashed lines indicate training onset and offset. **D**: Recurrent spiking network with variants of iSTDP (I-to-E plasticity only) in the familiarity task, with different Hebbian windows. Simulations were repeated 5 times, averages across seeds are shown in bold, dashed lines denote the last timestep at which the population firing rate in response to familiar stimuli was significantly different to novel responses (Student t-test, p < 0.05).

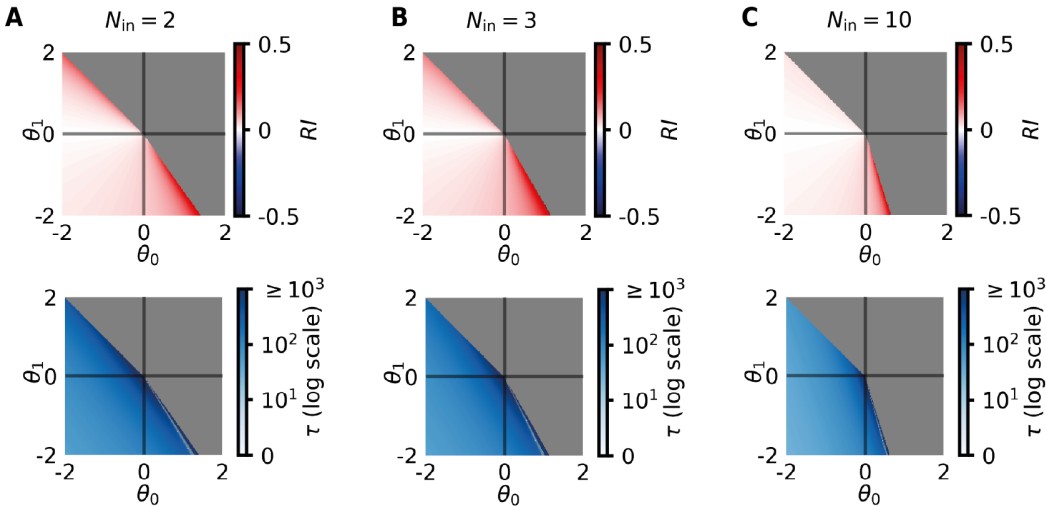

Supplementary Figure S8: **Extending the explicit toy model to higher dimensions.** **A**: Top: Phase portrait of the relative improvement $RI$ as a function of the values of $\theta_0$ and $\theta_1$. Note that here, $RI$ is defined as in appendix A.3.2, to extend to $N_{\text{in}} > 2$ input dimensions. Grey denotes unstable rules. Bottom: Same parameter sweep as C, but plotting the time to convergence $\tau$. These two plots are generated for $N_{\text{in}} = 2$ (same as main). **B,C**: Same as A, but for $N_{\text{in}} = 3$ and $N_{\text{in}} = 10$.

