# OpenReview forum: "Memory by accident: a theory of learning as a byproduct of network stabilization"
_NeurIPS.cc/2025/Conference — NeurIPS 2025 poster_

### Official Review · Reviewer_cFwp · 2025-06-15

**Clarity:** 3
**Significance:** 3
**Originality:** 4
**Rating:** 5
**Confidence:** 4

**Summary:**

This paper formalizes and analyzes the intriguing observation that stabilization of neural networks leads to memory, even in the absence of explicit memory mechanisms. This 'memory-by-accident' phenomenon is widespread and arises from the degeneracy of stabilizing weight matrices (i.e., the network state is history-dependent).

**Questions:**

The paper doesn't define stability or how the networks are optimized for stability.

Relatedly, one point I'm confused about is the relationship between stability optimization and the familiarity detection task. Were the networks trained to do the task, or optimize stability, or both?

The paper mentions a firing rate set point, but I didn't see where that parameter shows up in the model (sorry if I missed it).

p. 6: "the model predicted that the only rules that do not elicit memory are exclusively non-Hebbian". It's not clear to me that such a strong statement is warranted. I believe this is true for the toy model, but I don't think the authors have shown this in a more general form. There could exist non-Hebbian rules that elicit memory under other modeling assumptions.

I'd like to understand better what this means for neuroscience. Are there testable predictions? Does it change how we think about memory systems in the brain?

**Ethical Concerns:**

["NO or VERY MINOR ethics concerns only"]

**Final Justification:**

The authors have done a good job addressing my questions. I'm going to maintain my score of 5, because I think that this accurately reflects my current evaluation of the paper.

**Limitations:**

Yes

**Quality:**

4

**Strengths And Weaknesses:**

Strengths:
- The paper explores an interesting and provocative idea.
- Overall well-written.
- Mathematically rigorous.
- Opens up a new area of research.

Weaknesses:
- A few aspects need further explanation.
- Not entirely clear what the implications are for neuroscience.

---

> ### Author Rebuttal · Authors · 2025-07-30
>
> We thank the reviewer for their positive review and interesting suggestions, please find below detailed answers to all the points raised above.
>
> **1. “The paper doesn't define stability or how the networks are optimized for stability.”**
> This is indeed an oversight on our end, also raised by reviewer jDbn, which we fix in the revised version of the paper.
> In the recurrent spiking model, the definition of stability comes from the paper that inspired this work [1]: a network is stable if its activity and weight dynamics are biologically plausible for a wide range of inputs. I.e., activities remain within plausible ranges (non-silent network, not too-high firing rates, asynchronous irregular activity), as well as slowly changing weights that don’t reach extreme values. In the toy models, we somewhat narrow this description (L154-158) as both theoretical analysis (eqs.8 & 24) and simulations [1] show that the plasticity rules from eq.1 can be seen as establishing a setpoint on the network activity. We thus only consider rules with the property of bringing the neuronal activity to a setpoint and call them “rules that enforce network stability”. In this paper, saying that “networks are optimized for stability” is an abuse of notation on our end, we mean that the unsupervised plasticity rules embedded in the network can be seen as stabilizing network dynamics, i.e. bring the neuron activities to a firing rate setpoint. We have fixed and clarified these points in the revised paper.
>
> **2. Were the networks trained to do the task, or optimize stability, or both?**
> Since we are using unsupervised plasticity rules, the networks are not explicitly trained for anything, they receive inputs as part of the familiarity task and react to these inputs depending on the plasticity rule(s) embedded in the network. However, analysis performed on such rules here and in previous work [2, 3] shows these unsupervised rules can be understood as an approximation of gradient descent on a stability objective (forcing the network activity to a setpoint, see Supp L849). We have clarified this point in the revised version of the paper.
>
> **3. The paper mentions a firing rate set point, but I didn't see where that parameter shows up in the model**:
> We acknowledge that the several similar yet slightly different plasticity parameterizations introduced in the paper can be somewhat confusing. We have attempted to clarify and contrast these parameterizations in the revised manuscript.
> For the plasticity rules in the spiking network models (eq.1), the firing rate setpoint can be computed for each synapse type independently (eqs.8 & 24, L233) and depends on the plasticity parameters and the network activity. For the toy model with the explicit parameterization, the firing rate set point $ y^{\ast} $ is imposed by construction of the plasticity search space, we only consider rules of the shape eq.2 that admit $ y^{\ast} $ as a fixed point for all inputs of unit norm. For the implicit parameterization, the setpoint $ y^{\ast} $ is also imposed by construction, and we define the rules by the distance metric that they will minimize when bringing network activity to that setpoint.
>
> **4. p. 6: "the model predicted that the only rules that do not elicit memory are exclusively non-Hebbian. It's not clear to me that such a strong statement is warranted. I believe this is true for the toy model, but I don't think the authors have shown this in a more general form."**
> The reviewer rightfully points out that we don’t consider all possible rules, and as such we cannot prove that statement conclusively. Hence, we have rephrased this sentence to dial down and clarify this point, and added a point in discussion highlighting the limitation that there are many Hebbian/non-Hebbian rules that exist and that were not considered in our analysis. Nevertheless, we think that the evidence presented here across a variety of rule types/network models is sufficient to propose the theory of memory by accident despite this limitation.
>
> **5. “I'd like to understand better what this means for neuroscience. Are there testable predictions? Does it change how we think about memory systems in the brain?”**
> Besides the points raised in the discussion which are not necessarily directly testable predictions, we believe the impact of this work is two-fold:
> - 1/ Reinterpreting old data: Numerous evidence exists for Hebbian/anti-Hebbian plasticity in the brain [4,5]. However, previous theoretical work suggested that such simple rules are not enough to understand the memory abilities of brains [6,7]. Here, we propose that the evidence required to understand basic memory abilities in the brain may have been available for nearly 30 years. And we speculate in discussion that more complex memory abilities could be compositions using these simple building blocks. As such, we believe that memory-by-accident has interesting implications for neuroscience.
> - 2/ Testable predictions: our toy models formulated a few predictions that we tested *in silico* in the final figure, that link properties of the plasticity rule (synaptic mechanisms) and corresponding memory features (behaviour/cognitive features). For *in vivo* predictions, we could speculate that brain regions endowed with long-term storage might possess different learning rules (the ones suited for long memories, for instance with pre-only terms) than regions usually associated with faster forgetting such as potentially working memory, though all these rules may belong to the same Hebbian/non-Hebbian family of rules like the ones studied in this paper.
>
> **References**
> [1] Confavreux et al. biorXiv 2025
> [2] Vogels, Sprekeler et al. Science 2011
> [3] Kempter et al. PRE 1999
> [4] Markram et al. Science 1997
> [5] Bi and Poo, Journal of neuroscience 1998
> [6] Morrison, Diesmann and Gerstner Biological cybernetics 2008
> [7] Zenke, Hennequin and Gerstner PLoS Comp Bio 2013

---

> > ### Comment · Reviewer_cFwp · 2025-08-01
> > **Response to rebuttal**
> >
> > I thank the authors, who have done a good job responding to my comments. The only place where I feel the response falls somewhat short is the neuroscience implications; these are still quite vague and generic. That being said, I don't think it's critical for the authors to address this point.

---

> > > ### Author Response · Authors · 2025-08-08
> > >
> > > We thank the reviewer for encouraging us to clarify the implications of our work. Below we elaborate on the theoretical significance and specific predictions that emerge from our findings. We will add these points to the discussion of the revised paper.
> > >
> > > **1. Theoretical Implications**
> > >
> > > - Our work suggests that simple memory phenomena such as familiarity detection, novelty responses, and forms of statistical learning can emerge not from specialized memory mechanisms but as a byproduct of network stabilization via unsupervised plasticity. This reframes the longstanding search for purpose-built memory rules [Abbott & Nelson, 2000].
> > >
> > > - Degeneracy supports memory: Stabilizing plasticity rules select among many weight configurations that preserve activity dynamics. The selected configuration reflects prior inputs, resulting in activity-silent memory traces, which have been observed experimentally [Stokes et al., 2015].
> > >
> > > - Co-activity enables robustness: Rules that are unstable or uninformative in isolation can yield robust and long-lasting memories when co-active across synapse types (Fig. 2A and Fig.S1), highlighting the limits of analyzing plasticity in isolation [Zenke et al., 2015, Confavreux et al., 2025].
> > >
> > > - System size amplifies memory: Memory lifetimes increase with network complexity (network size, cell-types, number of co-active plasticity rules, Fig.2 and Confavreux et al. 2025). While a single rule in a small feedforward model may yield memories lasting for minutes, co-active rules in large recurrent networks can support hours-long memories (Fig. 2) and more complex behaviors like replay, sequence learning and omission novelty [Confavreux et al., 2025].
> > >
> > > **2. Experimental Predictions**
> > >
> > > *(a) Simple rules are sufficient for simple memories*
> > >
> > > We predict that if a neural system exhibits familiarity or novelty responses that outlast activity dynamics, synaptic plasticity is involved and simple (linear) combinations of Hebbian, pre-, and post-only terms are sufficient. This applies to a range of experimental observations from psychology, cognitive and systems neuroscience [Bogacz & Brown, 2003; Standing 1973, Brady et al., 2008;Peters et al., 2022; Lim et al., 2015].
> > >
> > > To test this, one could record both activity and weight dynamics during a familiarity task. We predict long-lasting, activity-silent memory traces alongside stable firing rates before and after learning. Some of this has been observed in the brain already, but not coupled with weight recordings [Peters et al., 2022; Lim et al., 2015, Stokes et al., 2015]. Sadly, to our knowledge, such experiments remain out of reach in vivo across learning for now.
> > >
> > > Classic plasticity rules revisited: Hebbian and anti-Hebbian rules observed experimentally [Markram et al., 1997; Bi & Poo, 1998, D’Amour et al. 2015] fall under the umbrella of the memory by accident framework and should elicit memories when embedded in networks, potentially with other (unobserved) rules of the same family such that the combination creates stable dynamics.
> > >
> > > *(b) Homogenization of weights predicts forgetting*
> > >
> > > Our models predict two timescales for the weight dynamics: (i) a fast decay driven by firing rate mismatches compared to the setpoint, and (ii) a slower homogenization of weights, which better aligns with the memory lifetime (Fig. 2D, as noted by reviewer wTcy). Thus, tracking weight variance over time could serve as a marker of memory persistence in vivo.
> > >
> > > *(c) A unified mechanism for working and long-term memory?*
> > >
> > > We speculate that both activity-silent working memory and longer-term familiarity memory may arise from similar plasticity mechanisms (e.g., STDP), differentiated only by the shape of the rule. Rules with fast forgetting times (eg. Fig.5B top) may underpin short-term memory, while slower or co-active ones support persistent storage (Fig.2A & Fig.5B bottom). These mechanisms could act as precursors to consolidated memory systems, e.g., hippocampo-cortical transfer.

---

### Official Review · Reviewer_jDbn · 2025-06-26

**Clarity:** 2
**Significance:** 3
**Originality:** 3
**Rating:** 3
**Confidence:** 3

**Summary:**

In this study, authors address a specific type of memory which consists in distinguishing familiar and novel stimuli. Results suggest that training recurrent E-I networks with unsupervised plasticity rules that enforce stability of the network, such simple form of memory emerges a byproduct of learning. Authors showcase this phenomenon using different architectures of networks that, while not designed to have memory, they have them nonetheless when subject to unsupervised learning enforcing stability. A variety of network architectures is discussed, going from larger recurrent spiking networks, networks with inhibitory plasticity, and analytically tractable toy models which are small feedforward networks. Results suggest an interesting relationship between memory time scale and learning rate, or memory time scale and stability.

**Questions:**

1) Why would stability be required for memory formation? If we have a recurrent network that invariably goes into one single attractor state and its firing rate diverges there, this seems like a very strong memory in an unstable network. Can authors comment on that?

2) How exactly is degeneracy related to stability and memory formation?

3) What do coefficients \theta_j in Eq. 2 stand for?

4) What is the meaning of the result on Fig 5D, bottom?

5) Why using a feedforward toy model and not a recurrently connected one?

**Ethical Concerns:**

["NO or VERY MINOR ethics concerns only"]

**Final Justification:**

A number of clarifications were provided during the discussion period and some of the discussed issues were addressed in revision, which allowed me to increase the overall recommendation. In spite of these improvements, the take-home message of the paper is still confusing, as the presentation of results is confounding proofs with observations. For the sake of scientific integrity, it is essential that claims presented in the paper are proven or convincingly demonstrated, which, in my opinion, is not yet achieved in the present version of the paper.

**Limitations:**

Limitations are discussed with some clarity, however, there are limitations beyond those discussed in the Limitation section. One limitation that is not described is that the models only address similarity / novelty detection.

**Paper Formatting Concerns:**

No formatting concerns.

**Quality:**

2

**Strengths And Weaknesses:**

_Strengths_:
1) The research question is well motivated in the introduction, and the discussion is appealing to read.
2) Several ideas presented in the paper are really interesting and worth further investigation.

_Weaknesses_:
1) To some degree this paper still reads as a work in progress and requires careful clarification of main concepts as well as a clarification and straightforward description of methods. Most main concepts rely on a previous (unpublished and not peer-reviewed) work, and they need to be clarified in revision.
2) The paper uses several model architectures (large recurrent spiking networks, feedforward networks, minimal toy models) and it remains to some degree unclear and unproven how well results transfer from one architecture to another.

3) The definition of the problem has to be carefully revised. This namely includes specifying and emphasising that the narrative is about one specific and type of memory (novelty / familiarity).

4) Previous related results could be better recapitulated (in the introduction) and current paper's results could be better discussed with respect to relevant related literature (in discussion). The writing should reflect what is demonstrated by the current results, what is strongly suggested by the current results and what is merely suggested. Overgeneralising the findings that are mere observations from specific cases should not be presented as proven.

5) The degeneracy of connectivity weights is mentioned several times (including in the abstract) as something that warrants memory formation, but how degeneracy is related to memory formation is not clearly explained nor demonstrated. The information the reader gets is that the weights of the toy model are assumed to be degenerate. This seems insufficient evidence for systematically linking degeneracy of weights and memory formation.

---

> ### Author Rebuttal · Authors · 2025-07-30
>
> We thank the reviewer for the detailed feedback. We would first like to clear a potential misunderstanding when reading the reviewer’s summary: “Authors suggest that enforcing network stability gives networks that are also capable of memory, without explicitly training networks with any specific plasticity rule.” In this paper, though we consider a variety of local unsupervised plasticity rules, the network models always evolve with one instance of these rules (1 plasticity rule for the the toy models and feedforward spiking model, and 4 co-active rules—one per synapse type— in the recurrent spiking network model). Since the rules are unsupervised, we cannot write that the networks were “trained” on the familiarity task, but the plastic networks were shown stimuli as part of the familiarity detection task, such that their weights could keep an imprint (or not) of these specific stimuli.
>
> **Definitions and clarity, “the paper is overall confusing”**:
> We acknowledge that the various models and plasticity rules used in the study, used to convey the generality of our findings, may have obscured its message. We have rewritten parts of the introduction and results to define more clearly what we mean by “memory” and “stability” in this paper.
>
> - *Definition of memory*: Following previous work (Confavreux et al. 2025 [1]), we consider simple forms of memory: familiarity and novelty detection, which ask whether a network can remember whether an external stimulus has already been encountered in the past. In the spiking models, the familiarity detection task was taken from [1]. We consider a network with weights plastic following a given plasticity rule or quadruplet of rules. We present the network once with a few stimuli (the will-be familiar stimuli). After the stimulus presentation is over, the network continues to evolve with its own internal dynamics while receiving background inputs. We wait for various durations ranging from 1s to 4h before doing a “test session” that aims to probe whether the network remembers the familiar stimulus or not. That means freezing the weights, and showing the network the familiar stimuli and novel stimuli in succession to obtain the mean firing rate of excitatory neurons during familiar stimulus presentation $r_{fam}$ and likewise for novel stimulus presentation ($r_{nov}$). We then define memory as the relative preference of the network for familiar or novel, $\Delta r_{mem} = 2(r_{nov} − r_{fam})/(r_{nov} + r_{fam})$. We hope this clarifies the current definition of memory that was L67-76 in the manuscript. We would be keen to hear any other suggestions the review may have as to what else we should measure to complement this definition. For the toy models, we adapted this definition due to the simplification of the network and of the task (described L159-175).
> *Why is this definition of memory relevant?* For the spiking network models, we refer the reviewer to the methods of [1] which builds on previous studies and provides numerous controls that this task does provide a simple and efficient definition of memory. First, the familiar stimuli are always compared to novel stimuli, such that the metric for memory reflects how different the network response is for familiar vs novel. This metric is shown to correctly flag networks without ongoing plasticity as not having memorized the familiar stimuli. Moreover, most rule quadruplets obtained in [1] forget after various timescales, which suggest that the metric is correctly flagging an effect specific to the familiar stimulus that is ultimately forgotten by ongoing network activity and plasticity.
>
> - *Definition of stability*: In the recurrent spiking model, the definition of stability comes again from the paper that inspired this work [1]: a network is stable if its activity and weight dynamics remain "biologically plausible" for a wide range of inputs, despite the ongoing plasticity in the recurrent synapses. I.e., activities remain within plausible ranges (non-silent network, not too-high firing rates, asynchronous irregular activity), as well as slowly changing weights that don’t reach extreme values.
> In the toy models, we somewhat narrow this description (L154-158) as both theoretical analysis (eq. 8) and simulations [1] show that the plasticity rules from eq.1 can be seen as establishing a setpoint on the network activity. We thus only consider rules with the property of bringing the neuronal activity to a setpoint $y^{*}$ and call them “rules that enforce network stability”.
>
> **1. Why would stability be required for memory formation?**
> We don’t see stability as a requirement for memory formation, but as a general observation in non-epileptic subjects and thus as a general requirement for models of the brain circuits and synaptic plasticity. The main conclusion of this paper is that stability enforced by simple local unsupervised rules tends to generate simple forms of memory as a byproduct. In the example provided by the reviewer, the divergence of the firing rate would preclude that network from being considered stable. Also, because the attractor state is not stimulus-specific, that network would not be flagged as memorizing the familiar stimulus.
>
> **2. How exactly is degeneracy related to stability and memory formation?**
> We observe that degeneracy is a consequence of stability (defined as bringing neuron activities to a setpoint): there are more weights than neurons in the network, so many weight combinations that will bring neurons to a setpoint value (L104-108, L140-142). We also observe that this degeneracy of weight matrices that can stabilize a network means that which weight matrix is currently implemented in the network can reflect previous inputs to the network which is a potential form of memory. In practice we verify this is the case for most local unsupervised rules we test in this study, at least transiently. These observations in the spiking models (fig 2B&C) become mathematically tractable and easy to visualize in the toy model (fig 3C for instance).
>
> **3. What do coefficients $\theta_j$ in Eq. 2 stand for?**
> The four thetas are four scalar parameters that determine the plasticity rule in the first toy model (L156). They determine to what extent weight updates depend neither on the pre- or the postsynaptic activity $\theta_0$, the presynaptic activity only $\theta_1$, postsynaptic activity only $\theta_2$ or both $\theta_3$ (i.e. a Hebbian term).
>
> **4. What is the meaning of the result on Fig 5D, bottom?**
> In Fig. 5D we take 20 “rule quadruplets” from [1], i.e. 20 vectors of plasticity parameters that each determine four plasticity rules of type eq.1. We know from [1] that these 20 quadruplets all create recurrent spiking networks with stable activity and weight dynamics (which we verify again in simulations). 10 of these quadruplets create memories with short memory lifetimes, $\Delta r_{mem} \neq 0$ only for a few seconds, whereas the other 10 have longer memory lifetimes (>4h). We ask how close these 20 quadruplets are to instability, i.e. what is the nearest quadruplet of rules that creates an unstable network. To do so numerically, we perturb the plasticity parameters for each quadruplet across random dimensions, with perturbations of increasing magnitudes (from 1 to 5). We report which perturbations generated stable networks and which ones did not, as a function of the magnitude of the perturbation, and plot for each quadruplet what was the closest unstable perturbation. However, the results do not show a significant difference between the quadruplets with short memories and those with long memories (L222-230).
>
> **5. Why using a feedforward toy model and not a recurrently connected one?**
> In this paper, we want to show a very general finding about memory and stability, and thus a result applicable both in feedforward and recurrent settings. We designed the toy model as the simplest possible model able to exhibit memory by accident: two weights, one postsynaptic neuron. We have also made a few recurrent toy models, mainly to study how the co-activity of several plasticity rules in the same network affects memory by accident (see fig S1). We refer the reviewer to our rebuttal for review zVuV, which discusses co-activity. In short, the problem is that many rules create unstable networks when embedded in isolation in the recurrent models, and require other rules to be stable. This complicates testing what each rule is good at. The feedforward models, both rate and spiking, do not have that problem and were thus a great way to decouple the stability problems and the memory capabilities of each rule. We mention this problem only briefly currently (L235-236), we have changed the text to elaborate on this more in parts 3.1 and 3.2, as well as in the discussion.
>
> **Reference**
> [1] Confavreux, Harrington, Kania, Ramesh, Krouglova, Bozelos, Macke, Saxe, Goncalves, and Vogels, Memory by a thousand rules: Automated discovery of multi-type plasticity rules reveals variety
> degeneracy at the heart of learning, bioRxiv, 2025

---

> > ### Comment · Reviewer_jDbn · 2025-08-03
> >
> > I thank the authors for replying to my questions, which indeed clarified some of several claims made in the paper. A fundamental question remains in what sense these networks have a memory of a stimulus. To clarify this further, I have a couple of follow-up questions.
> >
> > **Definitions and clarity**
> > 1) If the network was simply given two stimuli, A and B, it would likely converge to a different activity set points even without being previously presented with one of these stimuli. So it still remains unclear to me (and I think also in general) in what way does the novel / familiar setup that is used in this paper help understanding the phenomenon of memorising information by the brain networks.
> >
> > I would be better convinced that the network's activity has a memory of the stimulus if the stimulus could be decoded from network's activity, ideally in the absence of the stimulus. Or alternatively, if some information about the stimulus would be measured from network's activity (again, best in absence of the stimulus or in presence of partial information about the stimulus).
> >
> > 2) In Figure 2A, top, authors show a plot of the firing rate with three traces, "excitatory", "novel" and "familiar". Does the response to "novel" and "familiar" not include excitatory neurons and is thus the response of inhibitory neurons?
> >
> > 3) After training with STDP rules, the firing rate for novel stimuli is higher than for the familiar. Is this systematic (response to novel always higher compared to familiar) or do authors report a specific example? Which type of neurons (excitatory or inhibitory) increases / decreases the firing rate upon the presentation of a familiar stimulus? What is the reason for the observed decrease in the firing rate?
> >
> > 4) If the network was to modify its synapses in order to recognise a familiar stimulus upon its next appearance, would it make sense that the network would increase the firing rate to a familiar stimulus (to signal the detection) rather than decreasing it? A decrease in firing rate upon the presentation of a familiar stimulus seems to be better conceptualized as adaptation to the stimulus rather than memory.
> >
> > **Degeneracy**
> > In the abstract, there is the following claim: "We show that memory arises from the degeneracy of weight matrices that stabilize a network". This is a strong claim and it does not seem to be justified by the results, but seems to rather be an observation. Why not presenting it as an observation?
> >
> > **New question**
> > Does any of the results require networks to be spiking or could the same conclusions be achieved by rate networks? Can authors specify, which of the results explicitly require a spiking network?

---

> > > ### Author Response · Authors · 2025-08-04
> > >
> > > We thank the reviewer for taking the time to elaborate on their concerns. We hope that we have now better understood said concerns, and address them below. (Due to character limits, we have broken our response into 3 comments).
> > >
> > > **Definitions and clarity**
> > >
> > > 1. Do our networks exhibit memory of past stimuli?
> > >
> > >   (i) We employ a broader definition of memory in this paper than associative memories as they are usually defined for instance in Hopfield networks. Instead, we focus on familiarity/novelty detection: “the ability to discriminate between the relative familiarity or novelty of stimuli” [1], which has a long history in Psychology, Experimental and Computational Neuroscience [1-4]. For instance, this form of memory does not require pattern completion or the existence of one attractor state in the network per memory stored—in fact, our network models exhibit activity-silent memory (Fig.2, L95)—. All we need is “can the subject report whether the stimulus presented is familiar or novel?”, which corresponds for a network model to “can we decode whether a stimulus is familiar or novel from the network activity?”
> > >
> > >
> > >   (ii) Now that our definition of memory in the context of familiarity detection is clear, we provide additional details on the familiarity task that show how we test whether networks express a memory of previous stimuli (originally L754-762, we have rephrased and added the clarifications below).
> > >
> > > The network receives inputs from 11025 Poisson neurons arranged in a 2D grid. These input neurons are connected to the recurrent neurons with a non-random pattern (“receptive-field-like”, as in previous work [5,6]): for each recurrent neuron, we select a random input neuron as the center of the circular receptive field of radius 8.
> > > At baseline (no specific stimulus active), all the input neurons fire at 10Hz. When a stimulus is active (novel or familiar), 10% of the input neurons elevate their firing rate to 60Hz. There is no overlap across input stimuli, i.e. input neurons belong to at most one stimulus. In this paper, to make sure the results are robust, we use 7 stimuli (3 novel and 4 familiar).
> > > With this design, each stimulus activates some recurrent neurons more than others (namely the recurrent neurons that have in their receptive field input neurons that are part of the active stimulus). Each stimulus elicits a different network response, in the sense that each stimulus preferentially excites a different subset of recurrent neurons (this can be seen in FigS2) and thus the stimulus *identity* can be decoded from the population vector of neuron activities. However, whether this stimulus has been encountered by the network in the past —its *novelty* or *familiarity*— is unknown.
> > >
> > > We choose a very simple decoding strategy for stimulus familiarity: the mean firing rate of the excitatory population (each excitatory neuron contributes equally to the decoding). In the paper that inspired this work [5], a Student t-test is performed over several seeds/simulations/stimuli over mean firing rates in response to familiar or novel stimuli to determine whether novel stimuli elicit statistically different mean firing rates than familiar stimuli. Note that by design, all stimuli elicit statistically indistinguishable mean firing rates in static or naive plastic networks [5]. Thus this task does flag a stimulus-specific change in network activity due to synaptic plasticity.
> > >
> > > Overall, we would like to thank the reviewer for pushing us to add all the relevant information in the paper and not overly rely on assumptions and definitions from previous work. It seems that the term “memory” is used in the field with increasingly varied definitions. Although it is not this paper’s aim to compare and contrast all the definitions of memory currently in use in Computational Neuroscience, we hope the clarifications above are acceptable and that our goals and scope are now more clear.
> > >
> > >
> > >
> > > 2. The label “excitatory” refers to the mean firing rate of the excitatory population in the absence of specific input stimuli, whereas “novel” (resp. “familiar”) refer to the firing rate of the excitatory neurons during the presentation of a novel (resp. familiar) stimulus. Though this is explained in the figure legend, we agree with the reviewer that the labels can be confusing, and have now replaced “excitatory” with “baseline”.

---

> ### Author Response · Authors · 2025-08-04
>
> 3. We refer the reviewer to the original study from which this rule is taken from [5], in which a variety of behaviours are reported, not only novelty preference. Depending on the plasticity rule parameterization, the most widespread behaviour can be novelty or familiar preference [5]. For the simple Hebbian-non-Hebbian rules considered in eq.1, novelty preference is the most typical behavior observed, although the rule quadruplet chosen in Fig.2 creates exceptionally long and robust memories. Though we mention that this rule quadruplet is hand-picked from the dataset in [5] (L87-88), we have added another sentence at the end of this part saying that the effects are rule-dependent. It is precisely because we observe a variety of behaviors in the complex 4-rule-recurrent-spiking model that we attempt to simplify the problem with subsequent network models, culminating with the toy model.
>
> The excitatory neurons decrease their firing rates upon presentation of the familiar stimulus (or rather don’t increase it as much compared to a novel stimulus), the metric $\Delta r_{mem}$ does not consider inhibitory neurons, although controls in the original paper shows effects are similar [5]. It is complicated to pinpoint a single reason why the rates change, as 4 rules are cooperating in the network, but an explanation focusing on IE plasticity is developed in sections 3.2 and 3.3: IE weights onto the excitatory recurrent neurons that are excited by the familiar stimulus potentiate during the training period. During testing, the novel stimulus elicits a strong network response as the IE weights onto the recurrent neurons that are most excited by this novel stimulus have not been specifically potentiated during training, whereas the the familiar stimulus triggers a weaker response as the IE weights onto the recurrent neurons most excited by the familiar stimulus are still higher than others because of the past exposure to the familiar stimulus (Fig.2C & D).
>
> 4. There is experimental evidence for both behaviors (familiarity and novelty detection [7,8]), depending on the brain region and task. Though the rule quadruplet we focus on in Fig.2 displays novelty preference, similar rules in [5] do elicit the behaviour mentioned by the reviewer, following a similar memory-by-accident mechanism. Adaptation to a stimulus is a form of familiarity memory, though to the best of our knowledge, “adaptation” typically refers to shorter durations than behavioural timescale, hence the term “novelty detection” also in use [8]. Here, we happily write that the networks shown in Fig 2 display adaptation to the familiar stimuli on the timescales of minutes to hours, depending on the plasticity rule.
>
> **Degeneracy**
> We agree with the reviewer that the current statement is too general and can be clarified: we propose adding “familiarity detection memory using local, unsupervised Hebbian-non-Hebbian rules”, we are not claiming that all memories in the brain have to be stored using activity-silent memory and weight degeneracy, simply the memories studied in this article, for the class of rules of interest.
> However, besides this clarification, we believe that our observation of the degeneracy of weights during the memory task (shown in Fig.2B,C & D) is a proof of the link between the two. Indeed, the timepoints (1) and (3) show weight degeneracy: at these two timepoints, the network has the same activity setpoint in the presence of background inputs (Fig.2A top, red line), yet the weight matrices at (1) and (3) are different (Fig.2A middle/bottom, Fig.2B). Since the network responses at (3) to novel and familiar stimuli are significantly different, but not at (1), the network at (3) has a memory of the familiar stimuli at (3) but not at (1). Because the network model used has no other possible ways to remember past information besides its activity or weights, and since the memory is not in the activity (same activity setpoint at (1) and (3)), the imprint of the familiar stimulus has to be in the weights, and linked to the degeneracy observed.

---

> > ### Author Response · Authors · 2025-08-04
> >
> > **Spiking vs Rate**
> > Here, we show that the general phenomenon of memory by accident is widely applicable in rate or spiking networks, that can be feedforward or recurrent. However, in practice, two points lead us to believe that this phenomenon is most interesting in large recurrent spiking networks:
> > - The rule quadruplets obtained by meta-learning in the recurrent spiking networks did not translate into a rate equivalent (at least when using classical mean-field methods). In other words, rules that were stable in a large recurrent spiking network created runaway dynamics when embedded in a recurrent rate model (Fig.S1), or when embedded in isolation in recurrent spiking networks. Thus rules with interesting memory properties that should be unstable in practice can exist in recurrent spiking networks when working in concert with other rules. We only note this effect in the current paper and do not provide an explanation for it. We refer the reviewer to our rebuttal to reviewer zVuv (section on co-activity) that for more details.
> >
> > - For memory by accident to be able to create memories that are robust and long-lasting, the system needs to be complex enough. We believe this is why this phenomenon was not noticed or considered as memory-worthy before the meta-learning study of co-active rules in large recurrent spiking networks [5].
> >
> > **References**
> > [1] Bogacz and Brown, Comparison of computational models of familiarity discrimination in the perirhinal cortex. Hippocampus, 2003.
> > [2] Standing, Learning 10000 pictures, The Quarterly journal of experimental psychology, 1973.
> > [3] Brady, Konkle, Alvarez, and Oliva, Visual long-term memory has a massive storage capacity for object details.  Proceedings of the National Academy of Sciences, 2008
> > [4] Tyulmankov, Yang and Abbott, Meta-learning synaptic plasticity and memory addressing for continual familiarity detection, Neuron, 2022.
> > [5] Confavreux, Harrington, Kania, Ramesh, Krouglova, Bozelos, Macke, Saxe, Goncalves, and Vogels, Memory by a thousand rules: Automated discovery of multi-type plasticity rules reveals variety degeneracy at the heart of learning, bioRxiv, 2025.
> > [6] Zenke, Agnes, and Gerstner. Diverse synaptic plasticity mechanisms orchestrated to form and retrieve memories in spiking neural networks. Nature communications, 2015.
> > [7] Peters, Marica, Fabre, Harris, and Carandini, Visuomotor learning promotes visually evoked activity in the medial prefrontal cortex, Cell Reports, 2022.
> > [8] Lim, et al. Inferring learning rules from distributions of firing rates in cortical neurons, Nature Neuroscience, 2015.

---

> > > ### Comment · Reviewer_jDbn · 2025-08-04
> > >
> > > I thank the authors for their extensive reply and their patient explanation of key definitions and procedures that are used in the paper. While I understand that this work is tightly related to several previous studies (in particular the pre-print by Confavreux et al. 2025), I believe it is necessary to explain all important definitions in the present study to make it more clear and also more significant.
> > >
> > > Seen a number of clarifications that have been provided, and hopefully many of them will also be incorporated in the revised paper, I will increase my rating.

---

> > > > ### Author Response · Authors · 2025-08-06
> > > >
> > > > We thank the reviewer for their feedback and patience throughout the review process. We will ensure that all the clarifications discussed here are incorporated into the revised paper, so that it serves as a clear and fully self-contained account of the study.

---

### Official Review · Reviewer_ZVuV · 2025-06-30

**Clarity:** 4
**Significance:** 3
**Originality:** 3
**Rating:** 4
**Confidence:** 5

**Summary:**

In a preprint (cited) the group of Tim Vogels in Austria has reported a large simulation stuy where many combinations of STDP rules (E to E, E to I, I to I, I to E) yield network dynamics that are stable (in the sense that the  mean rate of excitatory neurons stays at, say 3Hz)  despite ongoing plasticity.

In the NeurIPS contribution, the authors develop a toy model to better understand this phenomenon. The toy model is a linear feedforward network with STDP rules translated into a rate model with Hebbian  term (preXpost)  as well as a pre-only and a post-only term.

The toy model works like this. Since the network is linear and (by construction) stable,  the postsynaptic neuron has a fixed point of the firing rate (say 3 Hz)  which translates into a linear manifold of weights compatible with that fixed point.
If a constant stimulus is presented, the weights move to a new linear manifold (consistent with the the same fixed point of the firing rate of 3Hz, but in presence of the stimulus). Once the stimulation is removed, the weights move back to the original manifold but end up at a different position. Therefore the weight vector contains information about the past stimulus that is not visible in the firing rate pattern.

**Questions:**

1. The authors illustrate the main idea for a toy model in two dimensions, but the math of linear manifolds could be and should be generalized to the case of N>2 weights, where some belong to the stimulated inputs part and others not. There should still be one Eigenvalue with a large negative real part (related to the stability of the fixed point), but within the group of stimulated neurons more eigenvalues will show up. What can we learn from these?

2. Kempter et al. 1999 have shown that in STDP rule where the Hebbian parameter gamma is nonzero, then even a purely Hebbian STDP rule induces indirectly a pre-only term (in the equivalent rate model) since every presynaptic spike influences the firing probability of the postsynaptic neuron proportionally to the weight: spike arrival at an excitatory/inhibitory synapse increase/decreases the postsynaptic rate which gives a correction term of order 1/N that should be included in the rate description that is used as rate equivalent of the original STDP model. This should be discussed.


3. Unfortunately,  we only learn at the very end (Fig 5B) that the strongest familiarization effect is the one with post-only (beta>0) in I to E connections. Moreover Fig S6 shows large heterogeneity of the weights. Both are interesting observation but deserve an explanation.

4. In what sense is the half-Hebbian rule FigS4D unstable? The firing rates show small fluctations but are stable over an hour. Weights transiently oscillate, but that does not indicate instability.

5. The analysis assumes that the fixed point is reached during the stimulation time. But obviously, if we turn down the learning rate a lot, this assumption is no longer correct. Also, given the rate stabilization dynamics, it should be straightforward to calculate the rate of convergence (i.e., time constant of stabiity eigenvalue) as a function of eta, and I would expect the authors to do this for both regimes: convergence time of the rate dynamics during stimulus presentation and without a stimulus.
My main concern here is that the statement that small eta helps to keep memory is both trivial and wrong. Trivial because weights will move slower. Wrong, since nothing will happen for eta too small. (Main prediction 2).

6. Main prediction 1: Is this related to a general result of dynamics and bifurcation theory? It is well known that at the transition from stable to unstable parameter regimes the (real part of) one of the Eigenvalues of the fixed point goes through zero.

**Ethical Concerns:**

["NO or VERY MINOR ethics concerns only"]

**Final Justification:**

I appreciate the good discussion with the authors. I feel that my concerns  regarding the limitations continue to be justified and I do not change my overall rating.

**Limitations:**

yes.

**Quality:**

3

**Strengths And Weaknesses:**

- Strengths

The paper is clearly written, starts with an interesting observation, and turns their observations into  a toy model. I like the analysis of the toy model. Even though the methodology is not novel (see Kempter et al. 1999, PRE, Kempter et al. 2001, Neural Computation) the application of the mathematical framework to the case of a familiarity memory after a short stimulion is new and interesting.

- Weaknesses

The abstract (as well as line 265 of the main text) seems to hint at a result that increasing from one plasticity rule to four plasticity rules would increase the dimension of the manifold of parameter combinations consistent with the stable firing rate fixed point. Unfortunately, this is not shown (or not developped in the main text), even though a linear toy model for, say a recurrent network with 3 neurons (2 excitatory, 1 inhibitory neurons) driven by  3 further input neurons  could be analyzed in the same way as the toy model. I see steps in this direction in the SIMULATIONS of the toy model in supplementary, but it should be possible to perform mathematical ANALYSIS for these cases. The analysis should get give you a spectrum of Eigenvalues for convergence to the fixed points in both stimulated and unstimulated case - and these eigenvalues will tell you a lot. One can then translate each of the 3 input neurons into a group of 10 input neurons with the same stability and still do the analysis.

For the toy model it remains unclear to the reader whether it is meant to be valid for E-E connections as well as EI/IE connections or not. It seems that Eq. (3) is generic and applicable to all cases. However, a lot of the later analysis and simulations seems to be focused on variations of iSTDP (which is the case that has already been studied by Vogels et al. cited ref 19). The results could have a bigger impact if they are shown to  more generally applicable.

Two of the three predictions look kind of obvious for somebody who is familiar with linear stability analysis.

Overall, my impression is that the paper contains cool ideas that could be explored further to make a convincing package of results. As it stands, the paper does not exploit the full potential of its ideas. May be good strategy would be to work on the topic a bit longer and then submit to another upcoming venue.

---

> ### Author Rebuttal · Authors · 2025-07-30
>
> We thank the reviewer for their thorough and overall positive review of our manuscript. We are glad that they appreciated the toy models developed and the general direction of the project. Below are detailed answers to the points raised in the review.
>
> **1. Extending the model to higher dimensions**:
> We chose the smallest model possible for ease of visualization, but as pointed out by the reviewer, we do not mention higher input dimensions. We propose to add two supplementary figures on the toy models with N>2 input neurons and mention this generalization of the toy model in the main text. In our hands, the N>2 model does not change the predictions reported in the paper. Overall, we observe stronger memories as there are more degrees of freedom (weights) for the same (scalar) postsynaptic neuron activity that is brought to a setpoint by a single plasticity rule. This suggests that the larger the network, the stronger the memory-by-accident phenomenon (currently discussed L265, but we added a pointer to the N>2 model there). For more details on the N>2 case and for lack of space here, we refer the reviewer to our rebuttal to reviewer wTCy.
>
> **2. Citing and discussing Kempter et al. 1999 and 2001**:
> We apologize for this oversight, and now cite these papers. Concerning the indirect pre-only term in their derivation, we chose a simpler route for the rate equivalent of the spike-based rules which neglects the effect of all correlations between pre- and postsynaptic neurons. This assumption has been made in other studies about plasticity in spiking networks [1-3], and provides a reasonable prediction of the network activity [1]. We added a discussion point about this.
> We are currently working on additional corrections to this spike-to-rate derivation beyond those performed in Kempter et al. 1999 that may explain the stability through co-activity phenomena and other mismatches between theory and simulations noted in several previous studies. We believe this is most appropriate for a separate, dedicated technical paper, as the paper at hand has already been flagged as "confusing" by other reviewers due to the many models and rules considered.
>
> **3. “Unfortunately, we only learn at the very end (Fig 5B) that the strongest familiarization effect is the one with post-only in I to E connections. Moreover Fig S6 shows large heterogeneity of the weights. Both are interesting observations but deserve an explanation”**.
> This hypothesis-testing of spike-based rules resulted from our analysis of the toy model, hence its appearance only at the end of the paper. These non-zero pre-only rules are not stable in isolation in recurrent spiking networks [3]. Thus the idea of testing all the rules in isolation in the feedforward spiking model was not clear to us initially, but now appears somewhat trivial following the development of the memory by accident framework.
> We thank the reviewer for the interesting observation that weight heterogeneity appears to depend on the rule used in the feedforward networks. We have run additional simulations to make sure this observation is robust (it is!) and have added it in the revised paper. However, these findings do not translate easily to the recurrent spiking network with co-active rules. As most of these IE rules with non-zero pre-only terms are unstable in isolation, we remain cautious in linking this observation to memory-by-accident. Overall, we agree that at present we spend too little time analysing and discussing the results of Fig.5, which we correct in the revised version of the paper.
>
> **4. “In what sense is the half-Hebbian rule FigS4D unstable?"**
> The simulation in FigS4D is unstable because the weights slowly ramp up despite being exposed to constant inputs. They eventually reach extreme values if simulated for longer than 1h. We chose to keep the plotting consistent across all spiking simulations, but in that case we agree that it makes the weight instability tricky to see. We fixed this in the revised version.
>
> **5. Concerns on main prediction 2: learning rate and memory**
> We agree that our prediction on learning rates in the toy model (L186) is valid for small but non-zero learning rates and requires waiting long enough for the steady state to be reached. We have clarified this in the revised version. We also added a theory vs simulations time-to-convergence plot (Fig 3E, which is currently only simulations as rightfully pointed out). Moreover, we have added a supplementary control figure about time-to-convergence for stimulus vs background inputs. However, because of the definition of “stimulus” and “background” inputs (two vectors of unit norm, which one is “stimulus” and “background” is arbitrary), the results are similar. We have rephrased the verification of this prediction by checking that indeed when the learning rate is too small, learning stops, to show in the end more of a U-shaped curve (memory timescale as a function of learning rate value).
>
> While we agree with the reviewer that the current prediction in the toy model is quite simple as it stands (see below for a proposed extension), we don’t think this is the case for the spiking models, which is why we kept this prediction in the paper. For instance, without the weight degeneracy (or a rule that cannot exploit it such as the rule in fig 5C), the timescales of learning and forgetting are identical, and thus the learning rate doesn’t change the duration of the memory much. However, what we see in practice in the spiking models are two timescales for forgetting (L119-123). The weight degeneracy allows the network to land in a state where the postsynaptic activity is close to the target and the weights have the imprint of the familiar stimuli. The memory then seems to be forgotten by noise-driven fluctuations (fig 2C&D). These noise-driven fluctuations also depend on the learning rate of the rule, but can be of much longer timescale than learning driven by the activity mismatch of the postsynaptic neuron compared to the activity setpoint. We have added a discussion point of this interesting effect that will need to be studied more in future work.
>
> However, we agree that the prediction made by the toy model does not reflect this more interesting phenomenon, which is why we propose a simple extension to the toy model to be able to formulate the same prediction in a more grounded way: the addition of a diffusive process (noise). This allows to recast the relative improvement RI (a distance between starting and final weights) as a duration for the diffusive process to erase the memory, thus capturing the two-timescale-process seen in spiking networks. We thank the reviewer for pointing this out and believe that prediction 2 is now more legitimate and interesting.
>
> **6. Concerns on main prediction 1: memory strength and distance to instability**
> As pointed out by reviewer wTCy, not all the rules bordering instability are good at making memories (for instance only the top ones, b>0, in fig 4B). This means the relationship between RI and the eigenvalues of the system (neuron activities) is not direct. As such we don’t think this prediction is a trivial byproduct from bifurcation theory. But this is a very interesting idea that we look forward to exploring further in future work.
>
> **Additional points**:
>
> - *Absence of a toy model for co-activity of plasticity rules*:
> We agree that a toy model for co-activity is another standout from [3]. We have attempted —and so far mostly failed at— producing a satisfying recurrent toy model with co-active rules, as documented in Fig S1. The recurrent model we show in Supplementary with the mean-field version of the spiking rules (eq.1) does not reproduce any behaviors observed in the spiking networks. This is mainly due to stability mismatches: a vast majority co-active rule quadruplets that were stable in recurrent spiking models in [3] lead to runaway weight dynamics in their mean-field (rate) version for over 95% of rules (Fig.S1, L861), thus preventing us from drawing any conclusions on their memory abilities. We have developed a feedforward linear model with 2 co-active rules. This model shows that memory strength (RI) generally increases compared to the single-rule case. However, this model does not capture any of the stability phenomena (rules unstable in isolation can be stable through co-activity). Overall, we fully agree that this is a great avenue for future work. However, since memory by accident doesn’t require co-activity to function and is a key phenomenon in its own right, we believe it is worthy of being told to the community on its own. Besides, a toy model able to capture co-activity likely requires us to delve deeper into spike timing effects (see point above about Kempter et al.). We have thus chosen to keep this out of the current paper, which was already flagged by other reviewers as already having too many rules and models in its current state.
>
> - *Overfocus on IE plasticity and iSTDP when testing predictions from the toy model:*
> We used iSTDP as it is convenient to simulate (robust and stable) and commonly used in the field for network stabilization, but typically not for learning itself (e.g. Litwin-Kumar and Doiron 2014 in which it is used in tandem with EE plasticity). But it is true that our predictions are not specific to IE synapses, and the paper was unnecessarily focused on iSTDP, thank you for rightfully pointing this out, we have addressed this in the revised version. Ensuring stability with the other rules in isolation is a little more tricky, though we can test the predictions in the feedforward spiking model. We have added a panel in Fig.5 testing the predictions with EE (or EI/II) plasticity in the feedforward spiking case, and find that predictions hold for all synapse types.
>
> **References**
> [1] Vogels, Sprekeler et al. Science 2011
> [2] Confavreux et al. NeurIPS 2020
> [3] Confavreux et al. bioRxiv 2025

---

> > ### Comment · Reviewer_ZVuV · 2025-08-04
> > **response**
> >
> > Thanks for the response to my comments.

---

### Official Review · Reviewer_wTCy · 2025-07-02

**Clarity:** 4
**Significance:** 3
**Originality:** 4
**Rating:** 5
**Confidence:** 5

**Summary:**

The authors address a recent result whereby plasticity rules that ensure stability of firing rates also endow networks with memory of past stimuli. The main insight is that weights are degenerate, and hence for a given input, there is a set of weights that can cause firing rate to have a desired “output”. Because plasticity was designed for stability, it will lead the weights to an arbitrary point in this set. Switching an input (writing a memory) will change the set of allowed weights, and cause the weights to converge towards a point on this new set. In general, when switching back the weights are not expected to return to the original configuration, hence providing memory.
The authors demonstrate this concept in various simplified settings, and also show links between the timescale of emergent memories and the stability of the network.

**Questions:**

Results on gradient descent after zero loss (Blanc et al, COLT 2020; Li, Wang, Arora ICLR 2022) suggest that weights might drift on the allowed manifold over time, towards flat areas. Do you see evidence of this in your case?
Figure 1C caption typo “after a 4h”
What is the justification for freezing plasticity after the first stimuli in the familiarity task?
In Figure 2A, the weights are still evolving after training ends. Are weights frozen?
Figure 2B. Only W_IE seems to strengthen the familiar engram in point 2. And for W_EE it’s the opposite. The description in the main text (line 102-104) seems inconsistent with the figure.
Rules with the best RI are close to instability (Fig 3,4). The other direction does not hold – there are “bad” rules close to instability. Do you know how this generalizes in higher dimensions?
Related to the previous question – did you check the edge of stability for a simple linear model in high dimensions? There it should be easier to compute.

**Ethical Concerns:**

["NO or VERY MINOR ethics concerns only"]

**Final Justification:**

I am keeping my strong support for acceptance - 5.
I am open to discussions with other reviewers/AC on whether it should be a 6.

**Limitations:**

yes

**Quality:**

4

**Strengths And Weaknesses:**

Strengths:
This is a simple and elegant explanation of a seemingly strange observation.
The toy models help gain insight into the phenomenon.
The writing is mostly easy to follow, but see some questions below.

Weaknesses:
The toy models give some insights, but are limited to two dimensions. It would be useful to understand whether dimensionality changes anything qualitatively.

---

> ### Author Rebuttal · Authors · 2025-07-30
>
> We thank the reviewer for their very encouraging appraisal of our work and for their suggestions. Please find below a detailed answer to all points raised.
>
> **Extending the toy model to higher dimensions**:
> We chose the smallest toy model possible for ease of analysis and visualization, but as pointed out by the reviewer, the findings readily extend to higher dimensions ($N_{input}>2$ input neurons), even though we did not emphasize it in the submission. Note that we could also increase the number of postsynaptic neurons $N_{output}$. However, because of the shallow feedforward architecture, this would effectively correspond to $N_{output}$ uncoupled networks receiving the same inputs. For the explicit parameterization (Fig.3), we could increase the dimensionality of the plasticity parameter space, for instance by adding higher order terms in the Taylor expansion ($pre^2$, $post^2$). We have not done so in the current paper as the rules from the spiking simulations (eq.1) naturally translated into the search space defined in eq.2 (see eqs.8, 24 and [1]), but this is definitely an interesting avenue for future work.
>
> *Description of the $N_{input}>2$ toy model, explicit parameterization*:
> We consider $N_{input}$ input neurons $\mathbf{x}$ projecting on a single output neuron $y$ with weights $\mathbf{w}$: $y = \mathbf{w}^T \mathbf{x}$. We choose $\mathbf{x}$ to be of unit norm with nonnegative entries. The plasticity rules are the same as for the 2D case (eq.3): $\frac{\partial w_i(t)}{\partial t}=(y(t)-y^*)(\theta_0+\theta_1 x_i(t))$.
> This is an affine system of ODEs, which can be written in vector form as $\mathbf{\dot w} = A\mathbf{w} + \mathbf{b}$ with $A = \mathbf{x}(\theta_0 \mathbf{1} + \theta_1 \mathbf{x})^T$ and $\mathbf{b}=-y^{\ast}(\theta_0 \mathbf{1} + \theta_1 \mathbf{x})$, and $\mathbf{1}$ a $N_{input}$-dimensional vector of ones.
> For a constant input $\mathbf{x}$, this system is rank one, and the non-zero eigenvalue $\lambda = \theta_0 \sum_{i<N_{input}}{x_i} + \theta_1 \sum_{i<N_{input}}{x_i^2}=\theta_0 \sum_{i<N_{input}}{x_i} + \theta_1 $ for unitary norm inputs. This is a generalization of the derivation presented in Supplementary (eq.17, L807-828).
>
> Edge of stability: For the system to be stable, we need $\lambda<0$, which translates for unit-norm, entrywise nonnegative inputs to $\theta_0 \sqrt{N_{input}} + \theta_1 <0$ and $\theta_0 + \theta_1 <0$. Note that this is a generalization of the condition given L823 in Supplementary. We added this analysis in Supplementary, as well as a supplementary figure reproducing Fig 3D&E for 10 input neurons.
>
> *$N_{input}>2$ toy model, implicit parameterization*:
> This toy model has the same network architecture as above, but the number of input neurons does change the number of plasticity parameters, as the rules are this time parameterized by a distance metric of the Mahalanobis family (with covariance $\Sigma \in \mathbb{R}^{N_{input} \times N_{input}}$, see sections 4.2 in the main paper and A.4 in Supplementary). This leaves us with $\frac{N_{input}(N_{input}+1)}{2}$ plasticity parameters. We added in supplementary the 3D and 10D equivalent of Fig.4B, more precisely some 2D slices of these cases for visualization.
>
> Edge of stability: This corresponds to $\Sigma$ losing its positive semi-definite property (i.e. at least one eigenvalue becomes 0). In 2D, as pointed out by the reviewer, though the "best" rules (with strong memories) are close to instability, there are also “bad” rules (rules with weak memories) close to instability (Fig.4B). We verified that this property scales to higher dimensions (new supplementary figure), and clarified in the revised paper that this is a necessary though not sufficient condition for maximal memory. This may explain the initially inconclusive verification at scale of this prediction in Fig.5D.
>
> Overall, increasing the dimensionality of the toy models did not change qualitatively the findings reported in the paper.
>
> **Weight drift once on the solution manifold, similar to Blanc et al, COLT 2020; Li, Wang, Arora ICLR 2022**:
> We see evidence of weight drift for the spiking networks (Fig.2 and 5, perhaps most clearly visible in Fig.2D). However, it is unclear whether it is purely noise-driven perturbations or if there is a general direction to the drift. As far as we know, this is an open question in the field. We would guess that there is a direction and that the reviewer's hypothesis is true, as we notice most weights ultimately revert back to homogenous values, but we are not sure to what extent this is rule-dependent. Whether this corresponds to a flatter region of the loss, we don’t know for sure, but this is definitely a very interesting avenue for future research.
>
> **What is the justification for freezing plasticity after the first stimuli in the familiarity task?**
> This is a clarity mistake on our end which we have addressed in the revised version: weights are always plastic unless we are probing the familiar or the novel stimulus during a test session. The results don’t change too much if we let weights be plastic also during the test sessions, as the rules are overall slow. We chose to freeze the weights during the test session to be able to precisely monitor the forgetting of the familiar stimulus (memory timescale), for instance in Fig.2 and 5.
>
> **Minor comments and typos**:
> We adapted the description of the co-active spiking simulation (L102-104) to describe more precisely the effect on each synapse type.
>
> **Reference**
> [1] Confavreux, Harrington, Kania, Ramesh, Krouglova, Bozelos, Macke, Saxe, Goncalves, and Vogels, Memory by a thousand rules: Automated discovery of multi-type plasticity rules reveals variety degeneracy at the heart of learning, bioRxiv, 2025

---

> > ### Comment · Reviewer_wTCy · 2025-08-03
> >
> > I thank the authors for clarifying the various points in their rebuttal.

---

### Decision · Program_Chairs · 2025-09-17

**Decision:**

Accept (poster)

**Comment:**

While one reviewer was uncertain about acceptance, most reviewers agreed that the finding was interesting and worthy of publication, provided the paper was duly clarified.

Personally, I would also note that the proposed "inevitability activity-silent memory" has clear implications for neuroscience, considering the debate about how much working memory and short-term memory rely on such activity silent, synaptic mechanisms (https://pubmed.ncbi.nlm.nih.gov/26051384/).

After discussion with reviewers, I recommend acceptance, provided that the necessary clarifications and explanations are actually incorporated in the final paper.